# Regulating electrostatic phenomena by cationic polymer binder for scalable high-areal-capacity Li battery electrodes

Jung-Hui Kim [1,5], Kyung Min Lee[2,5], Ji Won Kim[3], Seong Hyeon Kweon[2], Hyun-Seok Moon[1], Taeeun Yim [3] ✉, Sang Kyu Kwak [4] ✉ & Sang-Young Lee [1] ✉

Despite the enormous interest in high-areal-capacity Li battery electrodes, their structural instability and nonuniform charge transfer have plagued practical application. Herein, we present a cationic semi-interpenetrating polymer network (c-IPN) binder strategy, with a focus on the regulation of electrostatic phenomena in electrodes. Compared to conventional neutral linear binders, the c-IPN suppresses solvent-drying-induced crack evolution of electrodes and improves the dispersion state of electrode components owing to its surface charge-driven electrostatic repulsion and mechanical toughness. The c-IPN immobilizes anions of liquid electrolytes inside the electrodes via electrostatic attraction, thereby facilitating $Li^+$ conduction and forming stable cathode−electrolyte interphases. Consequently, the c-IPN enables high-areal-capacity (up to 20 mAh $cm^{-2}$) cathodes with decent cyclability (capacity retention after 100 cycles = 82%) using commercial slurry-cast electrode fabrication, while fully utilizing the theoretical specific capacity of $LiNi_{0.8}Co_{0.1}Mn_{0.1}O_2$. Further, coupling of the c-IPN cathodes with Li-metal anodes yields double-stacked pouch-type cells with high energy content at 25 °C (376 Wh $kg_{cell}^{-1}$/ 1043 Wh $L_{cell}^{-1}$, estimated including packaging substances), demonstrating practical viability of the c-IPN binder for scalable high-areal-capacity electrodes.

The promising potential of forthcoming innovative electronics, electric vehicles, and grid-scale energy storage systems has spurred the unremitting pursuit of high-energy-density Li batteries[1,2]. Accordingly, in addition to the ever-continuing search for advanced electrode active materials, designing high-areal-capacity (C/A) electrodes has garnered attention as a facile and scalable approach to achieve this goal[3]. High-C/A electrodes increase the energy density of a cell without requiring the synthesis of new electrode active materials.

To achieve high-C/A electrodes (=areal-mass-loading (M/A) × specific capacity of electrode active materials ($C_{sp}$))[4], the M/A should be maximized while stably maintaining the $C_{sp}$. However, owing to the use of thick electrodes (physical issue) and nonuniform charge transfer throughout the electrodes (electrochemical issue), conventional electrodes cannot achieve this requirement[5]. Particularly, the drying of processing solvents, such as N-methyl pyrrolidone (NMP) and water, during the fabrication of thick electrodes often induces crack formation and delamination from metallic current collectors, thus limiting the increase in the M/A values[6,7]. Additionally, with an increase in the electrode thickness, charge transfer in electrode active materials tends to demonstrate uneven and sluggish reaction kinetics in the

[1]Department of Chemical and Biomolecular Engineering, Yonsei University, Seoul, Republic of Korea. [2]Department of Energy Engineering, School of Energy and Chemical Engineering, Ulsan National Institute of Science and Technology (UNIST), Ulsan, Republic of Korea. [3]Department of Chemistry, Incheon National University, Incheon, Republic of Korea. [4]Department of Chemical and Biological Engineering, Korea University, Seoul, Republic of Korea. [5]These authors contributed equally: Jung-Hui Kim, Kyung Min Lee. ✉e-mail: yte0102@inu.ac.kr; skkwak@korea.ac.kr; syleek@yonsei.ac.kr

through-thickness direction of the electrodes, resulting in the loss of the $C_{sp}$ values[8–10].

To resolve these problems, previous studies on high-$C/A$ electrodes have mostly relied on mechanical pressing[5] and manufacturing techniques[11–13] (including freeze-drying, infiltration, and phase inversion). However, despite improvements in the apparent $M/A$ values, the maintenance of the $C_{sp}$ values has received little attention, thus preventing the achievement of meaningful $C/A$ values. Furthermore, these physical architecture-driven approaches often require complicated and cost/time-consuming fabrication processes[13], making it difficult to translate the research outputs beyond the lab scale. Therefore, exploring a new material chemistry that can be compatible with commercial slurry-cast electrode fabrication is required to develop practically scalable high-$C/A$ electrodes.

Here, we present a class of cationic semi-interpenetrating polymer network (denoted as c-IPN) binder, which can be readily applied to slurry-cast electrodes, with a focus on the regulation of electrostatic phenomena in electrodes. The c-IPN consisted of a crosslinked cationic polymer network and linear polymer (shown in Fig. 1). As a model electrode active material for this approach, high-capacity Ni-rich layered transition metal oxide (LiNi$_{0.8}$Co$_{0.1}$Mn$_{0.1}$O$_2$, NCM811) particles were selected. Compared to neutral linear electrode binders commonly used to date, the c-IPN binder acted as a cationic polymer surfactant and enabled electrostatic repulsion with electrode components, thereby alleviating solvent evaporation-induced internal stress and improving the dispersion state of electrode slurries. This beneficial effect of the c-IPN binder on the electrode slurries, combined with the mechanical toughness of the IPN structure, suppressed the crack evolution of the dried electrodes, thus enabling the fabrication of high-$M/A$ electrodes (reaching 96 mg cm$^{-2}$ which corresponds to a $C/A$ of 20 mAh cm$^{-2}$) with uniform topologies. From an electrochemical point of view, the cationic groups of the c-IPN binder immobilized the anions of liquid electrolytes via electrostatic attraction, while simultaneously exhibiting an electrostatic repulsion with Li$^+$, thus enabling the prevalence and uniform distribution of freely mobile Li$^+$ throughout the c-IPN cathode and establishing a stable cathode–electrolyte interphase (CEI) on NCM811.

Owing to the c-IPN binder, the cathode achieved a high-$C/A$ (20 mAh cm$^{-2}$) and a decent cyclability, while fully utilizing the theoretical $C_{sp}$ (210 mAh g$_{NCM811}$$^{-1}$) of NCM811. The obtained high-$C/A$ cathode was assembled with a Li-metal anode to fabricate a double-stacked pouch-type full cell. Under this constrained cell configuration, the resultant full cell exhibited a high-energy-density (376 Wh kg$_{cell}$$^{-1}$/1043 Wh L$_{cell}$$^{-1}$, including packaging substances), which are well placed among previously reported pouch-type cells.

## Results

### Enabling high-energy-density Li-metal full cells using the c-IPN cathodes

The superiority of the c-IPN cathode (with high-$C/A$) over the control cathode (with low-$C/A$) is schematically represented in Fig. 1. A key-enabling technology of the c-IPN cathode is the regulation of the electrostatic phenomena, which plays a viable role in the fabrication of high-$M/A$ cathodes, promotion of Li$^+$ conduction, and utilization of NCM811 throughout the cathode.

The effect of the c-IPN cathode on the specific energy of the cells and $C_{sp}$ of NCM811 was investigated as a function of the $M/A$ (Fig. 2a and Supplementary Table 1). With an increase in the $M/A$, the specific energy of the cells increased until saturation at approximately 427 Wh kg$_{cell}$$^{-1}$ (here, packaging substances were excluded) at an $M/A$ of 86 mg cm$^{-2}$. Additionally, the $C_{sp}$ of NCM811 (~210 mAh g$_{NCM811}$$^{-1}$) was maintained over a wide range of $M/A$, indicating the efficient utilization of NCM811 even in thick c-IPN cathodes (details will be discussed in the following subsection). Consequently, the c-IPN cathode provided a $C/A$ of 20 mAh cm$^{-2}$, which is approximately five times higher than that of current commercial Li-ion battery (LIB) cathodes[3] (~4 mAh cm$^{-2}$) based on neutral linear polyvinylidene fluoride (PVDF) binders.

Several previous studies on high-$C/A$ electrodes have utilized coin-type cells, which are not sufficient to demonstrate the scalability of the electrodes (Supplementary Table 2). To explore the practical feasibility of the c-IPN cathode, a double-stacked pouch-type full cell was fabricated (Fig. 2b), in which a c-IPN cathode ($C/A$ = 18 mAh cm$^{-2}$) was prepared using a slurry-cast method (Supplementary Fig. 1). The X-ray microtomography image of the full cell confirmed the presence of a thick (350 μm) c-IPN cathode (Fig. 2c). The c-IPN cathode was coupled with a Li-metal anode (20 mAh cm$^{-2}$, negative to positive electrode ($N/P$) capacity ratio of 1.1) under a limited amount (electrolyte mass/cell capacity ($E/C$) ratio of 2.5 g Ah$^{-1}$)[14,15] of carbonate-based electrolytes. The fabricated cell yielded a $C_{sp}$ of 210 mAh g$_{NCM811}$$^{-1}$, indicating an almost full utilization of the theoretical $C_{sp}$[16] of NCM811 (Supplementary Fig. 2). Additionally, the specific energy and energy density of

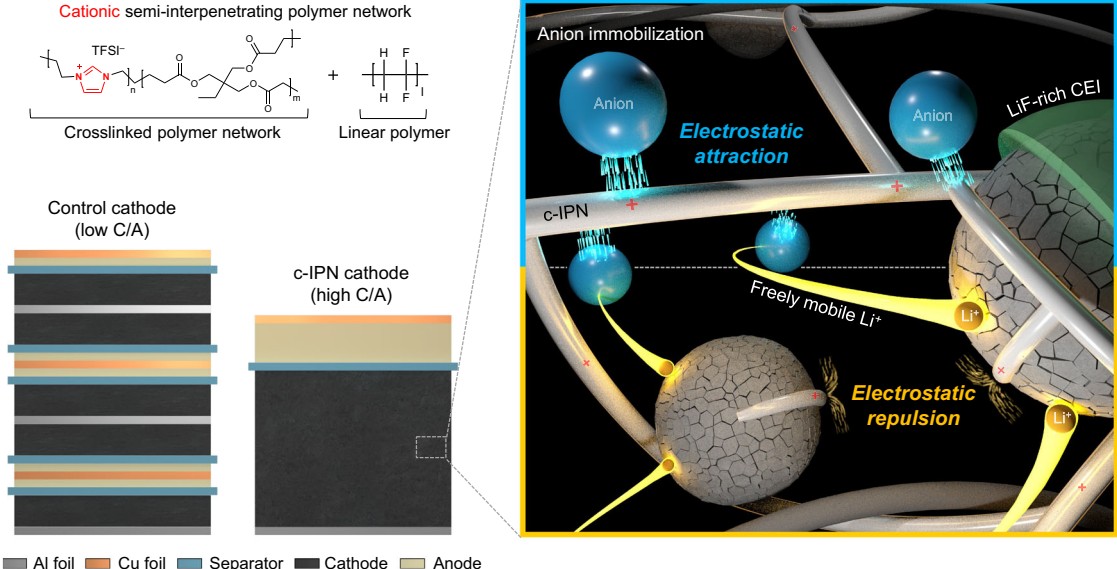

**Fig. 1 | Schematic illustration showing the superiority of the c-IPN cathode (with a high-areal-capacity ($C/A$)) over a control cathode (with a low-$C/A$) under the same cell capacity.** The regulation of the electrostatic phenomena by the c-IPN binder was depicted on the right.

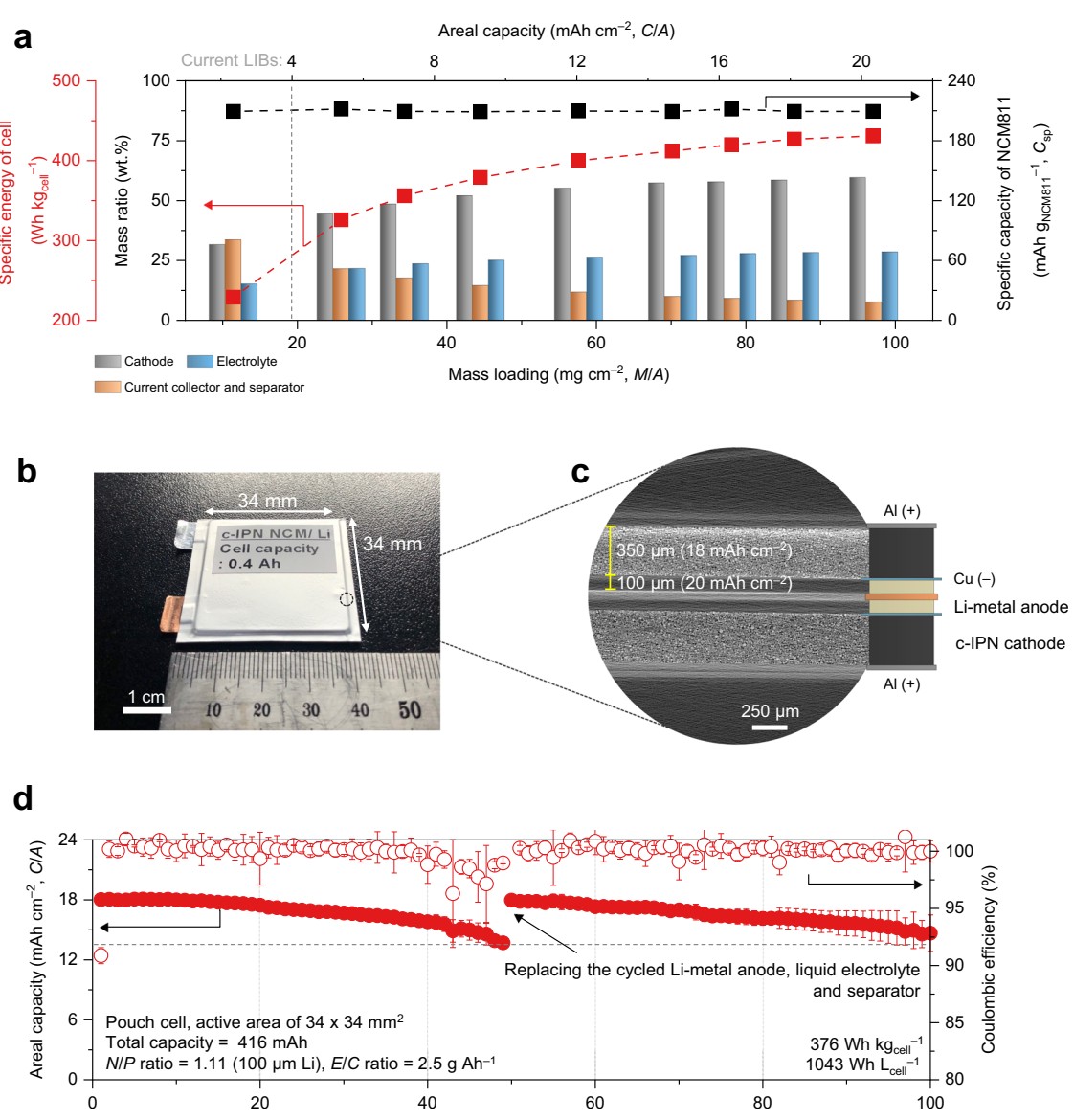

**Fig. 2 | Enabling high-energy-density Li-metal cells using the cationic semi-interpenetrating polymer network (c-IPN) cathodes. a** Effect of the c-IPN cathode on the specific energy of cells (excluding packaging substances) and the specific capacity of electrode active materials ($C_{sp}$) of NCM811 as a function of the areal-mass-loading ($M/A$). **b** Photograph of the double-stacked pouch-type cell (area = 34 × 34 (mm × mm)), in which the c-IPN cathode ($C/A$ = 18 mAh cm⁻²) was coupled with a Li-metal anode (20 mAh cm⁻², negative to positive electrode capacity ratio ($N/P$) of 1.1) under a limited amount (electrolyte mass/cell capacity ($E/C$) ratio of 2.5 g Ah⁻¹) of carbonate-based liquid electrolyte. **c** X-ray microtomography image showing the cross-sectional structure of the double-stacked pouch cell. **d** Cycling performance of the double-stacked pouch-type cell at a charge/discharge current density of 0.9 mA cm⁻²/1.8 mA cm⁻² and a voltage range of 3.0–4.4 V at 25 °C.

the cell was estimated to reach 376 Wh kg$_{cell}^{-1}$ and 1043 Wh L$_{cell}^{-1}$ (including packaging substances, Supplementary Table 3), respectively, which are well-placed among those of the previously reported high-energy-density pouch-type cells (Supplementary Fig. 3). We expect that the cell energy densities can be further increased by introducing a multicell stack configuration that will enable the reduction of the weight/volume portion of packaging substances in the full cells.

Moreover, the cell exhibited a stable cycling retention without the use of unconventional electrolytes that are specially designed for Li-metal anodes. Particularly, the cell almost completely recovered its initial $C/A$ after the replacement of the cycled Li-metal anode, liquid electrolyte, and separator with fresh ones (Fig. 2d and Supplementary Fig. 2), indicating that the c-IPN cathode is not the major cause of the

cycling decay. A supplementary experiment was conducted to further identify a major cause of this decline in the cycling retention (Supplementary Fig. 4). Replacing the cycled liquid electrolyte with a fresh one, while leaving the cycled Li-metal anode unchanged, failed to return to the initial voltage profile and showed rapid capacity fading with cycling. In comparison, replacing the cycled Li-metal anode with a fresh one returned to the normal and stable voltage profile. This result exhibits that the cycled Li-metal anode has a critical effect on the cycling degradation of the Li-metal full cell with the high-$C/A$ cathode. Thus, coupling the high-$C/A$ cathodes with electrochemically stable high-capacity anodes (such as advanced Li-metal or Si) should be conducted in future studies to highlight the cathode advancements for practical battery applications that require longer cycling retention and faster current rates.

## Regulation of the electrostatic phenomena for the dispersion state of c-IPN cathodes

The c-IPN consisted of PVDF and a crosslinked polymer network (cationic monomer (1-vinyl-3-allylimidazolium bis(trifluoromethanesulfonyl)imide, VAI-TFSI) and a crosslinking agent (trimethylolpropane triacrylate, TMPTA)) with a composition ratio of PVDF/(VAI-TFSI/TMPTA) = 60/(27/13) (w/w/w). Details on the synthesis of the VAI-TFSI are described in the Methods, along with the structural/electrochemical characterization of the crosslinked polymer networks and their optimal composition ratio (Supplementary Figs. 5–7). In addition to the c-IPN, PVDF and neutral semi-IPN (n-IPN, PVDF/TMPTA = 60/40 (w/w)) were selected as control binders. Three different cathodes consisting of the c-IPN, n-IPN, and PVDF binders were prepared at a composition ratio of NCM811/carbon black additive/binder = 92/3/5 (w/w/w), in which the thermal crosslinking of the c-IPN and n-IPN binders was conducted during the electrode fabrication.

The reduction of the surface tension of electrode slurries is known to alleviate drying-triggered internal stress which causes electrode cracking problem[17]. The c-IPN electrode slurry exhibited lower surface tension compared to the control electrode slurries (Fig. 3a and Supplementary Fig. 8), indicating that the c-IPN binder acted as an ionic surfactant. The presence of the cationic moiety in the c-IPN binder was identified by conducting zeta potential analysis. For this measurement, model suspensions (carbon conductive additive/binder = 3/5 (w/w) in NMP solvent) were prepared at a low concentration of 10 ppm, details of which are described in the Methods. The model suspension with the c-IPN binder showed a higher absolute zeta potential value (-14 mV) compared to the control suspensions with the neutral binders (-1 mV) (Fig. 3b), demonstrating the cationic nature of the c-IPN binder. It is known that the high absolute zeta potential value of a suspension solution is beneficial for improving the dispersion state[18]. Accordingly,

the c-IPN electrode slurry exhibited a good dispersion state, whereas some particle aggregates were observed in the control electrode slurries (Fig. 3c).

To investigate the effect of the electrode slurry states on the dispersion uniformity of the dried electrodes, laser scanning confocal microscopy (LSCM) analysis, which can map the surface roughness, was conducted. At a low-$M/A$ (e.g., 26 mg cm$^{-2}$), all the electrodes exhibited a uniform topography (Supplementary Fig. 9). However, with an increase in the $M/A$ to 50 mg cm$^{-2}$, the control electrodes exhibited severe deviation in height, indicating the nonuniform dispersion of the electrode components (Fig. 3d). In contrast, the deviation in the height of the c-IPN electrode over the wide range of the scanned area was negligible owing to its well-dispersed slurry state. This result demonstrates the viable role of the c-IPN binder for the regulation of the electrostatic phenomena in the slurry-cast electrode fabrication, which crucially affect high-$M/A$ electrodes.

### Structural stability of c-IPN cathodes

To elucidate the basic properties of the c-IPN binder, a self-standing c-IPN film was prepared through the thermal crosslinking of TMPTA and VAI-TFSI in the presence of PVDF. Peaks of TMTPA ($v_{C=O}$ and $v_{C-H}$) and VAI-TFSI ($v_{N-H}$ and $v_{C-F}$), along with disappearance of the characteristic peaks assigned to acrylic C = C bonds[19,20], were observed in the Fourier-transform infrared (FT-IR) spectroscopy results of the c-IPN film (Fig. 4a), indicating successful synthesis of the crosslinked copolymer network. Additionally, the phase-separated structure of the c-IPN, which is a typical feature of IPN[21], was verified by the detection of two individual glass transition temperatures (Supplementary Fig. 10).

The c-IPN and n-IPN films exhibited higher mechanical toughness (represented by the area in the stress–strain curves, Fig. 4b) than the PVDF film owing to their interpenetrating polymer networks[22,23].

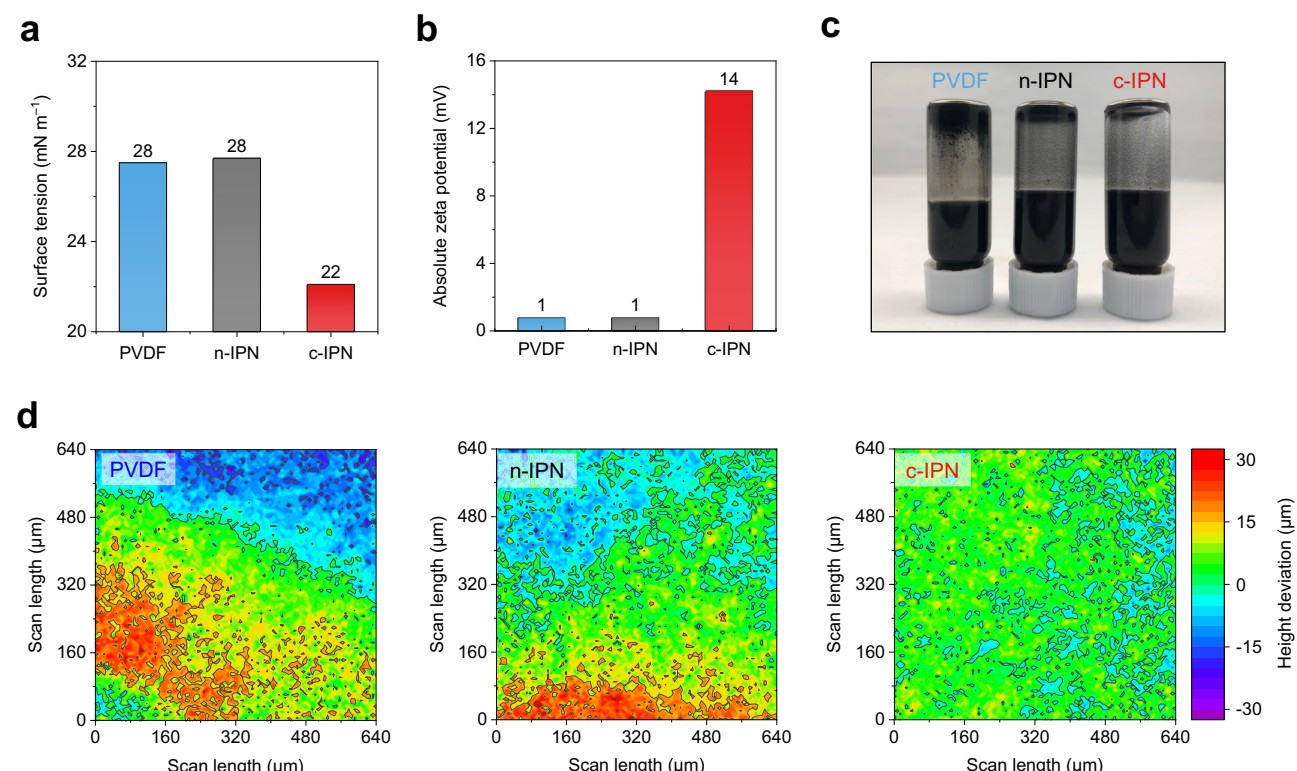

**Fig. 3 | Regulation of the electrostatic phenomena for the dispersion state of the c-IPN cathodes. a** Surface tension of the electrode slurries measured using pendant drop tensiometry. **b** Absolute zeta potential of the electrode slurries. **c** Photograph of the electrode slurries with a solid content of 30%. **d** Laser scanning confocal microscopy (LSCM) surface topography images of polyvinylidene fluoride (PVDF) (left), neutral semi-IPN (n-IPN) (middle), and c-IPN (right) electrodes with a high-$M/A$ ( = 50 mg cm$^{-2}$).

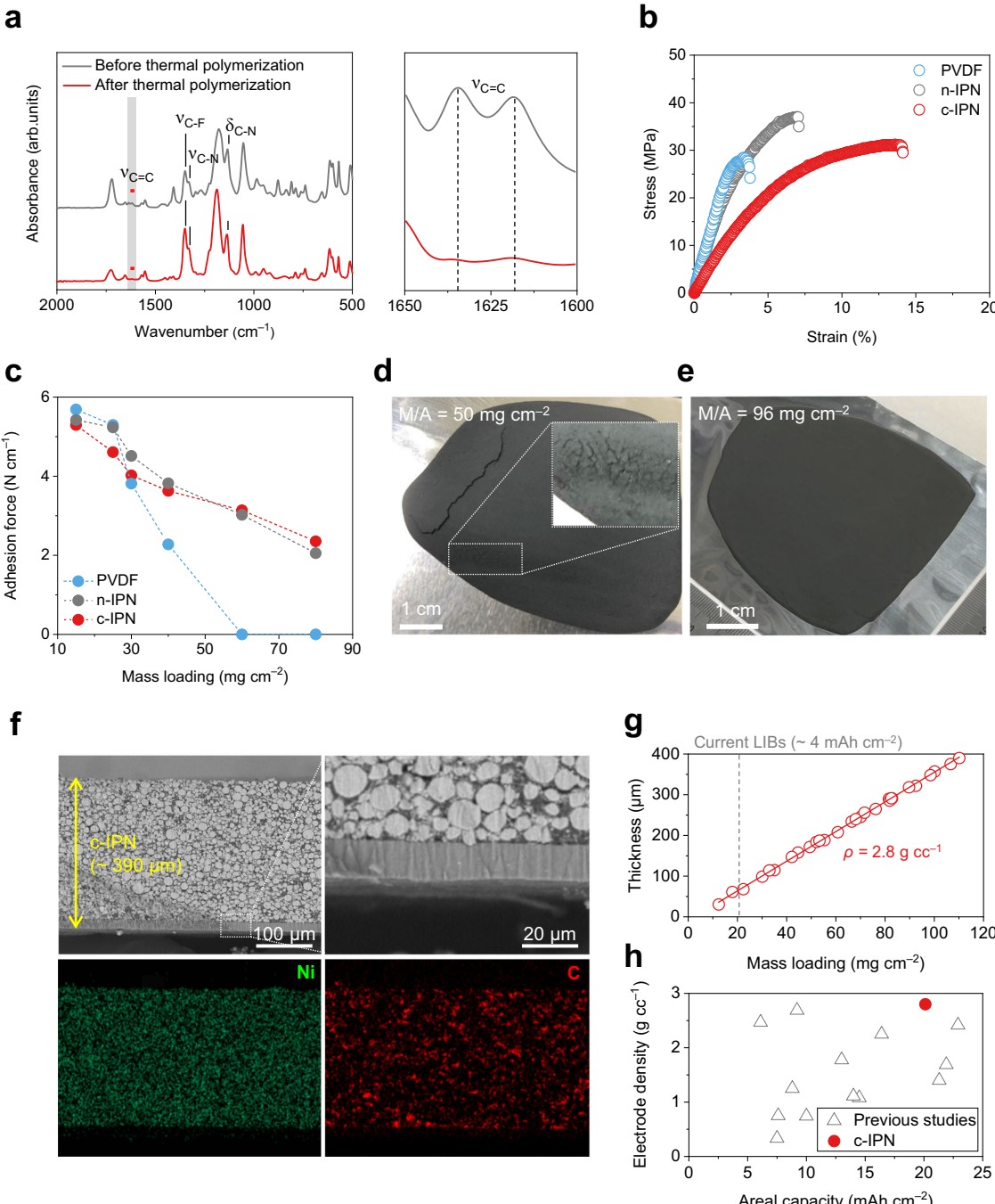

**Fig. 4 | Structural stability of c-IPN cathodes. a** Fourier transform infrared (FT-IR) spectra (focus on the characteristic peaks assigned to acrylic C = C bonds) of the c-IPN film before/after the thermal crosslinking. **b** Stress–strain curves of the PVDF, n-IPN, and c-IPN films. **c** Adhesion force (measured by 180° peel-off test) of the electrodes (PVDF vs. n-IPN vs. c-IPN) as a function of $M/A$. **d, e** Photographs of the PVDF ($M/A = 50$ mg cm$^{-2}$) **d** and c-IPN ($M/A = 96$ mg cm$^{-2}$) **e** electrodes. **f** Cross-sectional scanning electron microscopy (SEM) and energy dispersive X-ray spectroscopy (EDS) images of the c-IPN electrode ($M/A = 96$ mg cm$^{-2}$). **g** Thickness and electrode density ($\rho$) of the c-IPN electrode as a function of $M/A$ after the roll pressing. **h** Electrode density of various electrodes as a function of $C/A$: c-IPN electrode vs. previously reported high-$M/A$ electrodes.

Meanwhile, the higher toughness and lower elastic modulus of the c-IPN film compared to those of the n-IPN film may be attributed to loosely crosslinked networks formed by the copolymerization of TMPTA (a crosslinking agent with three double bonds) with VAI-TFSI (a monomer with a two double bond). This result indicates that the c-IPN binder can be mechanically tolerant against the solvent drying-triggered transverse internal stress[6,24] encountered during electrode fabrication.

The adhesion forces between the electrode active layers and Al current collectors were estimated as a function of the $M/A$. The adhesion forces tended to decrease as the $M/A$ increased (Fig. 4c and Supplementary Fig. 11). Particularly, at $M/A$ values above 30 mg cm$^{-2}$, the difference in the adhesion forces of the c-IPN (and control n-IPN) and PVDF electrodes was significant. Additionally, at an $M/A$ of 50 mg cm$^{-2}$, the PVDF electrode exhibited catastrophic cracking and delamination (Fig. 4d), which is consistent with previously reported

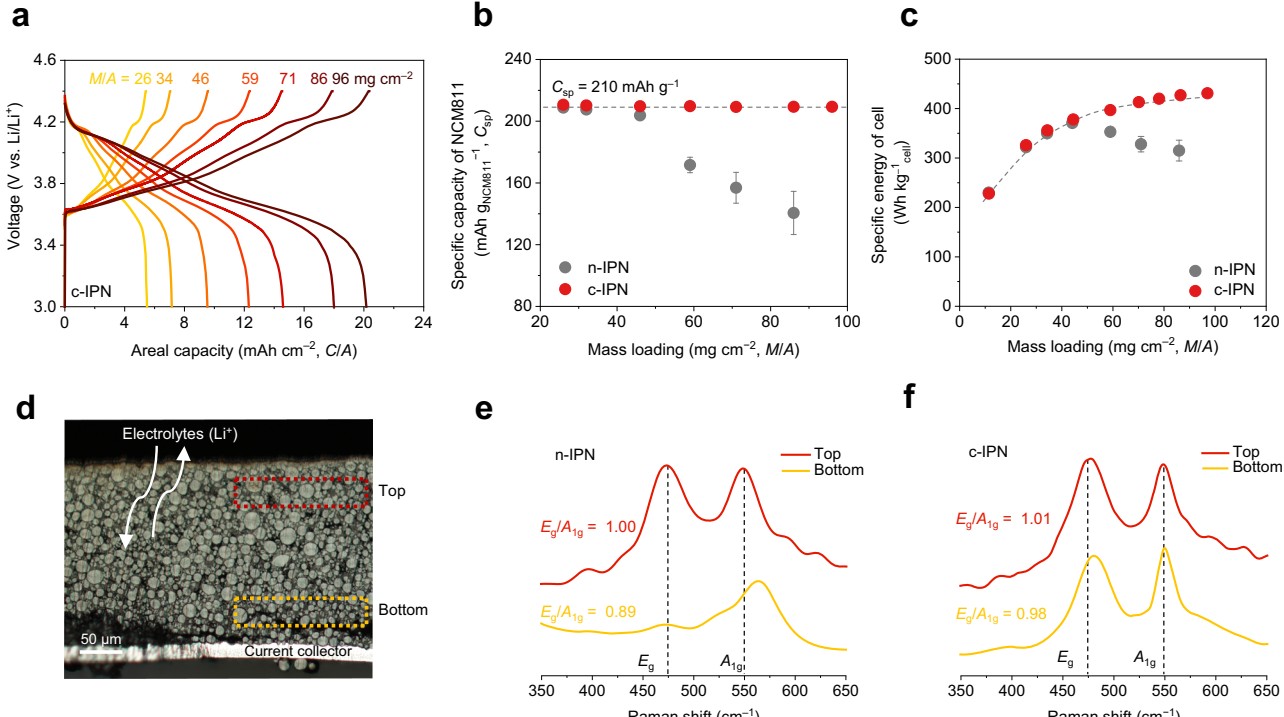

**Fig. 5 | Through-thickness redox uniformity of the c-IPN cathodes.**
**a** Galvanostatic charge/discharge profiles of the cells (c-IPN cathode ||Li-metal anode (100 μm, corresponding to $C/A$ of 20 mAh cm$^{-2}$)) as a function of the $M/A$ of the cathodes at 25 °C. **b** $C_{sp}$ of the c-IPN cathode (vs. control n-IPN cathode) as a function of the $M/A$ at discharge current rate of 0.1 C. The dotted line represents the theoretical $C_{sp}$ (~210 mAh g$^{-1}$) of the NCM811. **c** Specific energies (excluding the packaging substances) of the cells containing the c-IPN cathode (vs. n-IPN cathode) as a function of $M/A$. **d** Cross-sectional optical micrograph of the charged c-IPN cathode ($M/A$ = 65 mg cm$^{-2}$), in which the cells were charged to 4.4 V (corresponding to 100% SOC) at a current density of 1.1 mA cm$^{-2}$, and then disassembled to collect the charged cathode. The two boxes (indicating the top and bottom regions) in the image were selected for the Confocal Raman spectroscopy. **e**, **f** Raman spectra and intensity ratio of $E_g/A_{1g}$ of the n-IPN **e** and c-IPN **f** cathodes with $M/A$ of 65 mg cm$^{-2}$ at the top and bottom regions, in which $E_g$ and $A_{1g}$ represent the bending mode of metal-oxygen-metal in the $a/b$-axis direction and stretching mode of metal-oxygen complex in the $c$-axis direction, respectively.

results[3]. In contrast, the c-IPN electrode maintained its dimensional integrity without any cracks even at a higher $M/A$ of 96 mg cm$^{-2}$ (Fig. 4e). This result was verified by analyzing the cross-sectional structure of the c-IPN electrode at an M/A of 96 mg cm$^{-2}$ (Fig. 4f). The electrode active layer (thickness ~390 μm) of the c-IPN electrode exhibited close interfacial contact with the Al current collector. Moreover, the electrode components, in which Ni and C elements originate from NCM811 and carbon black additives/binders, respectively, were uniformly distributed in the through-thickness direction of the electrode. In addition to the aforementioned high-$M/A$, the c-IPN electrode exhibited a constant electrode density of 2.8 g cc$^{-1}$ over the entire $M/A$ range (Fig. 4g), which exceeded those of previously reported high-$M/A$ electrodes (Fig. 4h).

**Through-thickness redox uniformity of c-IPN cathodes: Maintaining the theoretical $C_{sp}$ of NCM811**
With increasing $M/A$, the $C/A$ of the c-IPN cathodes increased proportionally (Fig. 5a). Additionally, the cathode maintained the theoretical $C_{sp}$ (~210 mAh g$_{NCM811}^{-1}$) of NCM811 even at an $M/A$ of 96 mg cm$^{-2}$ (corresponding to a $C/A$ of 20 mAh cm$^{-2}$) (Fig. 5b), whereas the control n-IPN cathodes exhibited a sharp decay in the $C_{sp}$ at an $M/A$ of above 59 mg cm$^{-2}$ (Supplementary Fig. 12). The difference in the $C_{sp}$ of the c-IPN and control n-IPN cathodes was more noticeable at faster discharge current rates, as well as at higher $M/A$ values (Supplementary Fig. 13). Meanwhile, the PVDF cathode was excluded from this comparison because it could not be used to fabricate high-$M/A$ cathodes.

The ultimate goal of high-$M/A$ electrodes is to achieve high-energy-density (= voltage × $C/A$ ( = $M/A$ × $C_{sp}$)) cells. Several previous studies on high-$M/A$ electrodes have often neglected the significance of $C_{sp}$ (Supplementary Table 2). The c-IPN cathodes enabled a steady

increase in the specific energy of the cells with increasing $M/A$, which eventually reached a plateau of 431 Wh kg$_{cell}^{-1}$ at an $M/A$ of 96 mg cm$^{-2}$ (Fig. 5c and calculation details in Supplementary Table 1, in which packaging substances were excluded from the cell weight[6]). In contrast, with an increase in the $M/A$ beyond 59 mg cm$^{-2}$, the specific energy of the cells fabricated using the control n-IPN cathodes decreased, which is consistent with the $C_{sp}$ results (shown in Fig. 5b).

To further elucidate the difference in the $C_{sp}$ of the c-IPN and control n-IPN cathodes, their state of charge (SOC) was analyzed in the through-thickness direction. The local SOC values of the cathodes were examined as a function of the distance from the Al current collectors (denoted as "top" and "bottom") using Confocal Raman spectroscopy (Fig. 5d and Supplementary Fig. 14). It is known that the intensity ratio of $E_g/A_{1g}$ in the Raman spectra tends to increase with increasing SOC (i.e., degree of de-lithiation) of cathode active materials[25]. At 100% SOC, the control n-IPN cathode exhibited a lower $E_g/A_{1g}$ intensity ratio in the bottom region (Fig. 5e), indicating that the NCM811 does not easily undergo de-lithiation in the bottom region (adjacent to the Al current collector) owing to the uneven and tortuous pathway of Li$^+$ in the cathode. In contrast, the $E_g/A_{1g}$ intensity ratio of the c-IPN cathode remained almost unchanged regardless of its cross-sectional regions, verifying the through-thickness uniformity in the redox reaction of NCM811 (Fig. 5f). This redox uniformity of the c-IPN cathode was attributed to the facile Li$^+$ migration (rather than the electron conduction) in the through-thickness direction (Supplementary Fig. 15 and 16).

**Enhancement of Li$^+$ conduction inside cathodes by the c-IPN binders**
Binders are known to act as an adhesive between the cathode active materials (not completely covering them), while leaving a large portion

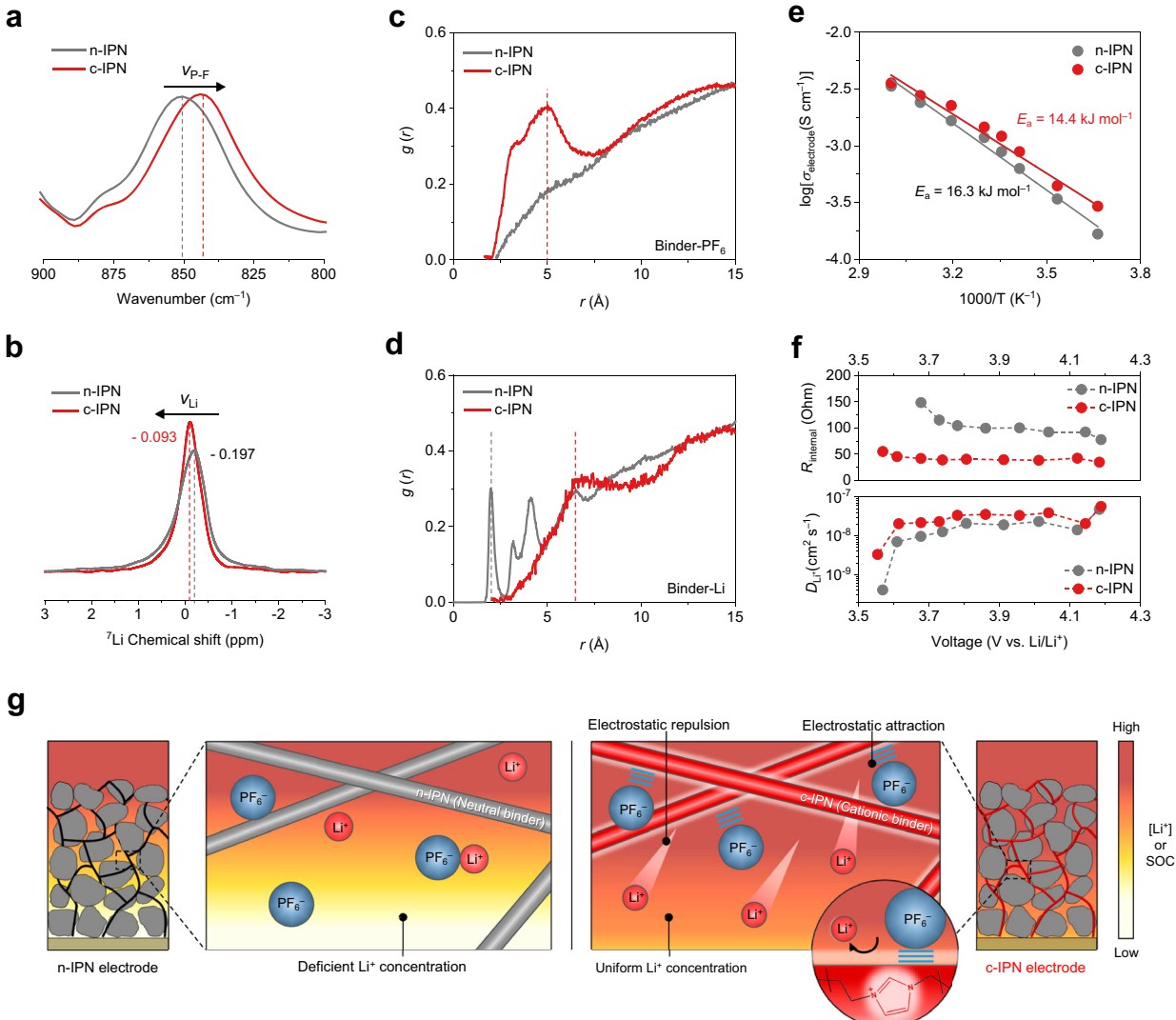

**Fig. 6 | Enhancement of the Li⁺ conduction inside the cathodes by the c-IPN binders. a** FT-IR spectra (focusing on the characteristic vibration peak (at 836 cm⁻¹) of P–F bonds) of the c-IPN (vs. control n-IPN) cathodes in contact with the liquid electrolytes (1 M LiPF₆ in EC/DEC = 1/1 (v/v)). **b** ⁷Li magic-angle spinning (MAS) nuclear magnetic resonance (NMR) spectroscopy results of the c-IPN (vs. control n-IPN) cathodes. **c, d** Radial distribution function (RDF) profiles of the P atom in PF₆⁻ **c**, and Li atom **d** at the binder–electrolyte interface. **e** Arrhenius plots for the ionic conductivities in the electrolyte-filled pores of the c-IPN (vs. control n-IPN) cathodes ($M/A = 65$ mg cm⁻²), which were obtained by the electrochemical impedance

spectroscopy (EIS) analysis of the symmetric cells (electrode | |electrode) at 0% SOC. **f** $R_{internal}$ and $D_{Li+}$ of the c-IPN (vs. control n-IPN) cathodes ($M/A = 65$ mg cm⁻²) as a function of the discharge voltage, which were estimated from the galvanostatic intermittent titration technique (GITT) results at 25 °C. **g** Schematic illustration showing the role of the c-IPN binder in enabling the prevalence of freely mobile Li⁺ with uniform distribution inside the cathode through electrostatic attraction with PF₆⁻ and electrostatic repulsion with Li⁺ in the liquid electrolyte.

of the cathode active material exposed to electrolytes[26]. Thus, the c-IPN binder could affect the surface charge environment in the electrolyte-filled interstitial pores of the c-IPN electrode, thereby contributing to the facile Li⁺ migration described above. The imidazolium-based cationic groups of the c-IPN binder enabled electrostatic attraction[19,27] with the anion (PF₆⁻) of liquid electrolyte (1 M LiPF₆ in ethyl carbonate (EC)/diethyl carbonate (DEC) = 1/1 (v/v)). Compared to that of the control n-IPN cathode, a downfield shift of the vibration peak (at 836 cm⁻¹) of P–F bonds was observed in the FT-IR spectrum of the c-IPN cathode (Fig. 6a), indicating the electrostatic interaction of PF₆⁻ in the liquid electrolyte with the positively-charged imidazolium moieties of the c-IPN binder. The local environment of Li⁺ in electrolyte-filled pores of the c-IPN cathode was identified using ⁷Li magic-angle spinning (MAS) nuclear magnetic resonance (NMR) spectroscopy (Fig. 6b). Compared to the control n-IPN cathode, the c-IPN cathode exhibited a downfield shift and a narrower width in the

⁷Li spectrum, indicating the improvement in the dissociation of Li salts[28] and the mobility of free Li⁺[29], respectively. From the normalized intensities of ⁷Li-NMR spectra ($I/I_0$) plotted as a function of time, spin-lattice relaxation time ($T_1$) values were obtained (Supplementary Fig. 17). A smaller $T_1$ value is known to reflect faster diffusion rate of ions[30]. The c-IPN cathode showed a smaller $T_1$ value (737 ms) than the n-IPN cathode (957 ms), verifying the faster Li⁺ mobility in the electrolyte-filled pores.

The distribution of ions at the binder–electrolyte interface was investigated by analyzing the radial distribution function (RDF) of PF₆⁻ and Li⁺ (see the details in Methods and Supplementary Data 1-4). For the n-IPN binder, the RDF of PF₆⁻ increased gradually with distance ($r$) (i.e., distance from the n-IPN binder surface increased) (Fig. 6c). In contrast, the c-IPN binder exhibited a distinct peak at an $r$ of 5.0 Å, indicating that PF₆⁻ exists dominantly around the c-IPN surface. Subsequently, the RDF of Li⁺ was examined as a function of $r$. The c-IPN

binder exhibited a characteristic peak at a larger $r$ (6.5 Å) compared to the n-IPN binder (2.0 Å) (Fig. 6d), indicating that Li$^+$ is away from the c-IPN binder possibly owing to the electrostatic repulsion with the positively-charged imidazolium moieties of the c-IPN binder.

To examine the ion conductivity ($\sigma_{electrode}$) in the electrolyte-filled pores of electrodes, the electrochemical impedance spectroscopy (EIS) analysis of the symmetric cells was performed (Supplementary Fig. 18). A slope observed in the low frequency region of Nyquist plots is known to reflect ionic resistance in the electrolyte-filled pores of electrodes ($R_{ion}/3$, $\sigma_{electrode} = 1/R_{ion}$)[31]. The c-IPN cathode exhibited a higher $\sigma_{electrode}$ over a wide range of temperatures and a lower activation energy ($E_a$) compared to the control n-IPN cathode (Fig. 6e). This result was verified using galvanostatic intermittent titration technique (GITT) analysis (Supplementary Fig. 19). The cell with the c-IPN cathode exhibited lower internal resistances ($R_{internal}$) during discharge reactions (Fig. 6f). Additionally, the c-IPN cathode exhibited higher Li$^+$ diffusion coefficients ($D_{Li+}$) than the control n-IPN cathode over the entire voltage range (Supplementary Table 4). Meanwhile, the difference in the electrolyte uptake of the c-IPN and n-IPN cathodes was negligible (Supplementary Fig. 20), indicating that the facile Li$^+$ transport of the c-IPN cathode hardly depends on the amount of electrolyte inside the cathode. These results demonstrate that the c-IPN binder, driven by its electrostatic attraction with PF$_6^-$ and electrostatic repulsion with Li$^+$, enabled the prevalence of freely mobile Li$^+$ and uniform distribution of Li$^+$ concentration (Fig. 6g).

## Enhancement in the cycling performance by c-IPN cathodes and mechanistic understanding

The effect of the c-IPN cathodes on the cycling performance of Li-metal full cells (cathodes ($C/A = 13.5$ mAh cm$^{-2}$)||Li-metal anodes ($C/A = 20.0$ mAh cm$^{-2}$), N/P ratio = 1.5) was investigated. The cell with the c-IPN cathode exhibited a higher initial $C_{sp}$ (~210 mAh g$_{NCM811}^{-1}$) than that with the control n-IPN cathode (~ 185 mAh g$_{NCM811}^{-1}$) (Supplementary Fig. 21). Moreover, under this constrained cell condition, the c-IPN cathode exhibited a higher cycling retention than the control cathode (capacity retention after 100 cycles = 82% vs. 16% for the control n-IPN cathode) (Fig. 7a). To further elucidate the effect of the c-IPN cathode on the cycling performance, we analyzed the EIS spectra of the cells. The c-IPN cathode showed a lower cell resistance than the n-IPN cathode at the 1$^{st}$ cycle (Supplementary Fig. 22 and Supplementary Table 5). This result verifies the advantageous effect of the c-IPN binder on the prevalence of freely mobile Li$^+$ and the uniform distribution of Li$^+$ concentration in the high $C/A$ cathode (as shown in Fig. 6). Notably, the lower film resistance of the c-IPN cathode was observed after the 50$^{th}$ cycle, exhibiting the formation of stable CEI on NCM811 that could positively affect the cycling performance.

To investigate the CEI formed on NCM811, the NCM811 in the cycled cathodes were analyzed using X-ray photoelectron spectroscopy (XPS) depth profiles as a function of the etching time. The control n-IPN cathode exhibited a large content of organic elements (reflected by C and O) (Fig. 7b), which mostly originated from the decomposition of carbonate solvents in the electrolytes. In contrast, the c-IPN cathode exhibited a higher content of inorganic elements (such as Li and F) over a broad range of etching time, combined with the formation of LiF-rich components (Fig. 7c and Supplementary Fig. 23). It is known that the LiF-rich CEI layer tends to alleviate undesired interfacial side reactions with electrolytes, thus positively contributing to the cycle life[32,33]. This unique CEI structure of the c-IPN cathode was further characterized using the time-of-flight secondary ion mass spectrometry (TOF-SIMS) depth profiling (Supplementary Fig. 24), which is consistent with the result of LiF-rich CEI layer formed in the LiPF$_6$-based concentrated electrolytes[33].

The difference in the CEI layers of the c-IPN and n-IPN cathodes was further elucidated by investigating the Li$^+$ solvation structure of the liquid electrolyte inside the cathodes. Two characteristic peaks

assigned to the carbonyl stretching vibration of EC solvents in the liquid electrolyte were observed in the FT-IR spectrum of the n-IPN cathode at 1770 and 1800 cm$^{-1}$, which corresponded to Li$^+$-coordinated and free-state carbonates, respectively[19] (Fig. 7d). In the FT-IR spectrum of the c-IPN cathode, the peak at 1770 cm$^{-1}$ down-shifted with an increase in the intensity, indicating an increase in the Li$^+$-coordinated carbonates. In addition, the c-IPN cathode exhibited a relatively higher coordination number (CN) between Li$^+$ and solvents (Supplementary Fig. 25) and lower CN between Li$^+$ and PF$_6^-$ (Supplementary Fig. 26) in the liquid electrolyte, verifying the electrostatic attraction between PF$_6^-$ and the c-IPN binder.

The Li$^+$ solvation structure of the liquid electrolyte inside the cathodes was theoretically investigated using molecular dynamics (MD) simulations (see the details in Methods and Supplementary Table 6). Compared to the n-IPN cathode, the c-IPN cathode exhibited an increase in the ratio of Li$^+$-coordinated carbonates and a decrease in the ratio of undissociated LiPF$_6$-carbonate complexes (Fig. 7e and Supplementary Fig. 27). Additionally, we observed that the predominant Li$^+$ solvation structure inside the c-IPN cathode was the 5-0-0 configuration, implying that Li$^+$ is coordinated with five EC solvents, zero DEC solvent, and zero PF$_6^-$, whereas the common Li$^+$ solvation structure inside the n-IPN cathode is the 3-0-2 configuration.

The highest occupied molecular orbital (HOMO) energy levels of the major Li$^+$ solvation structures were investigated using density functional theory (DFT) calculation (Fig. 7f). Overall, the solvation structures of the c-IPN cathode exhibited lower HOMO energy levels than those of the n-IPN cathode, indicating the difficulties in the oxidation of the coordinated complexes close to the c-IPN binder. In the 5-0-1 Li$^+$ solvation structure, the c-IPN cathode exhibited a lower binding energy of Li$^+$ with PF$_6^-$ than the n-IPN cathode, indicating that PF$_6^-$ is weakly bound to Li$^+$ because of its electrostatic attraction with the c-IPN binder (Fig. 7g). Consequently, PF$_6^-$, which is adjacent to the c-IPN binder, was vulnerable to decomposition, thus promoting the formation of PF$_6^-$-derived LiF-rich CEI layers.

The structural change of the cycled NCM811 was analyzed using focused-ion-beam nanotomography (FIB) and high-resolution transmission electron microscopy (HR-TEM). In addition to the anisotropic strain-induced microcracks (indicated by red circles in Fig. 8a), the cycled NCM811 (n-IPN cathode) exhibited severe intergranular cracks along the grain boundaries[34]. In contrast, the cycled NCM811 (c-IPN cathode) maintained its structural integrity without noticeable cracks and disruption (Fig. 8b and Supplementary Fig. 28). Meanwhile, a thin and uniform CEI was observed on the cycled NCM811 (c-IPN cathode), whereas the cycled NCM811 (n-IPN cathode) exhibited thick and uneven CEI layers (Fig. 8c, d).

The layered structure of the cycled NCM811 in the n-IPN cathode was almost lost, which was confirmed by the corresponding Fast Fourier Transform (FFT) pattern (Fig. 8e, g) revealing an amorphous phase[35]. For the cycled NCM811 (c-IPN cathode), the initial layered structure (R$\overline{3}$m phase[34]) was stably maintained in the bulk region (indicated by Region #1 in Fig. 8f, h), while an NiO rock-salt structure (Fm$\overline{3}$m phase[34]) was observed in the thin outermost (~ 6 nm) region (Region #2). This structural stability of the cycled NCM811 (c-IPN cathode) could account for the capacity recovery of the double-stacked pouch-type cell (shown in Fig. 2d).

## Discussion

In this study, we demonstrated c-IPN as a class of cationic binder that can enable the fabrication of scalable high-$C/A$ electrodes. Compared to the typical neutral linear electrode binders, the mechanically tough c-IPN binder acted as a cationic surfactant and ensured an electrostatic repulsion with electrode components, thereby suppressing the solvent-drying-induced crack evolution and improving the dispersion state of the high-$M/A$ cathodes.

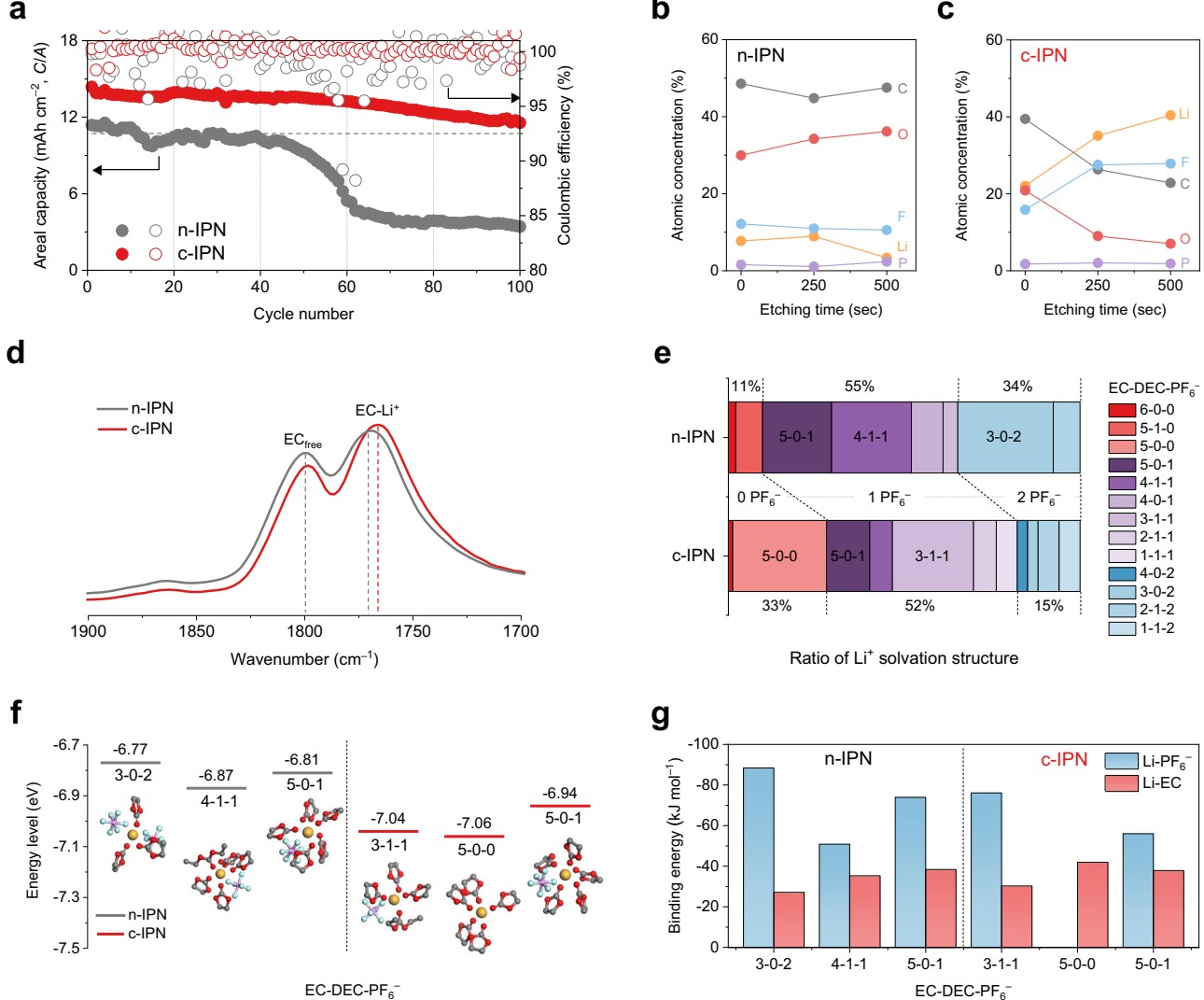

**Fig. 7 | Enhancement of the cycling performance by the c-IPN cathodes and mechanistic understanding. a** Cycling performance of the Li-metal full cells (n-IPN cathode vs. c-IPN cathode), in which the cathodes with an $M/A$ of 65.0 mg cm$^{-2}$ (corresponding to $C/A$ = 13.5 mAh cm$^{-2}$) were assembled with Li-metal anodes ($C/A$ = 20.0 mAh cm$^{-2}$, $N/P$ ratio = 1.5). The cells were cycled at a charge/discharge current density of 0.68/1.35 mA cm$^{-2}$ and a voltage range of 3.0–4.4 V at 25 °C. **b, c** X-ray photoelectron spectroscopy (XPS) depth profiles of the cycled n-IPN **b**, and c-IPN **c** cathodes. **d** FT-IR spectra (focusing on the characteristic peaks (at 1770 and 1800 cm$^{-1}$) assigned to carbonyl stretching vibration of the EC solvents in the liquid electrolyte (1 M LiPF$_6$ in EC/DEC = 1/1 (v/v)) of the n-IPN and c-IPN cathodes. **e** Relative ratio of the Li$^+$ solvation structure of the liquid electrolyte inside the n-IPN and c-IPN cathodes. The Li$^+$ solvation structures are denoted as *X-Y-Z*, where *X* is the number of EC molecules coordinated with Li$^+$, *Y* is the number of DEC molecules coordinated with Li$^+$, and *Z* is the number of PF$_6^-$ coordinated with Li$^+$. **f** Highest occupied molecular orbital (HOMO) energy levels of the major Li$^+$ solvation structures (n-IPN cathode vs. c-IPN cathode). **g** Binding energies of Li$^+$ with the electrolyte components (PF$_6^-$ and EC) in the major Li$^+$ solvation structures.

Additionally, the imidazolium-based cationic groups of the c-IPN binder immobilized the PF$_6^-$ of the liquid electrolyte and electrostatically repelled the Li$^+$ inside the cathode, thus ensuring the prevalence and uniform distribution of the freely mobile Li$^+$ throughout the c-IPN cathode and promoting the formation of LiF-rich stable CEI. Consequently, owing to the c-IPN binder, the fabricated cathode provided a high-$C/A$ (20 mAh cm$^{-2}$) along with a stable cycling retention, while fully utilizing the theoretical $C_{sp}$ (210 mAh g$_{NCM811}^{-1}$) of the NCM811. The high-$C/A$ c-IPN cathode was coupled with a Li-metal anode ($N/P$ ratio = 1.1) to fabricate a double-stacked pouch-type cell with high-energy-density (376 Wh kg$_{cell}^{-1}$/1043 Wh L$_{cell}^{-1}$). We envision that this cationic binder strategy based on the regulation of electrostatic phenomena is promising as a facile and versatile platform technology for the development of scalable high-$C/A$ electrodes, which cannot be easily achieved using conventional neutral linear binders.

## Methods

### Synthesis of 1-vinyl-3-allylimidazolium bis(trifluoromethanesulfonyl)imide (VAI-TFSI)

As an intermediate for the VAI-TFSI, 1-vinyl-3-allyl-imidazolium bromide (VAI-Br) was synthesized. 10.0 mmol of 1-vinylimidazole (Sigma-Aldrich) was mixed with 50 mL of acetonitrile (Sigma-Aldrich) and 12.0 mmol of allyl bromide (Sigma-Aldrich). Then, the acetonitrile was eliminated by rotary evaporator and the resulting crude product was purified by decantation using 20 mL of ethyl acetate and 20 mL of diethyl ether, respectively. The obtained 7.5 mmol of VAI-Br was dissolved in 50 mL of deionized water. Subsequently, 7.5 mmol of lithium bis(trifluoromethanesulfonyl)imide (LiTFSI) was added in the solution. The resulting mixture was then extracted using 100 mL of dichloromethane and was further purified by extraction with 50 mL of deionized water to remove organic impurities remaining in the solution. The obtained dichloromethane layer was purified by short-column

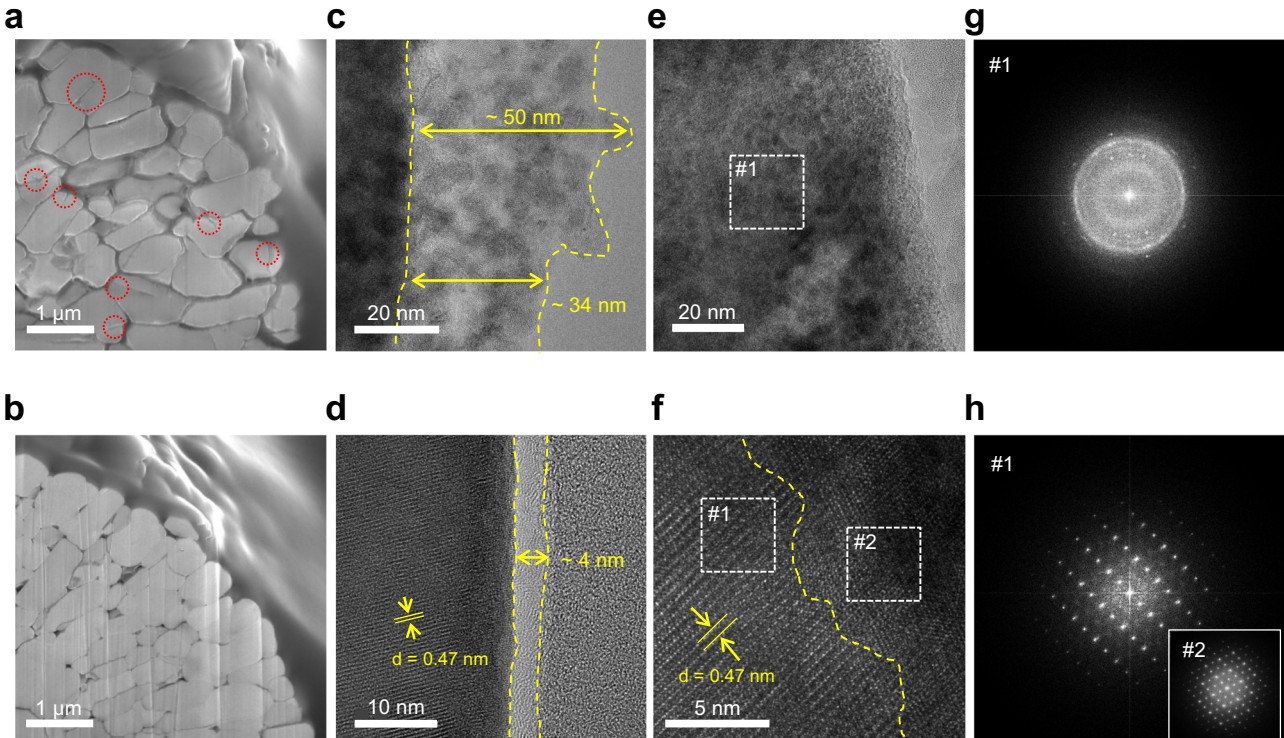

**Fig. 8 | Structural characterization of the cycled NCM811. a, b** Cross-sectional SEM images of the cycled NCM811 in the n-IPN **a** and c-IPN **b** cathodes. **c–f** High-resolution transmission electron microscopy (HR-TEM) images of the cycled NCM811 in the n-IPN **c, e** and c-IPN **d, f** cathodes. **g, h** Corresponding fast Fourier transform (FFT) patterns of the cycled NCM811 in the n-IPN **g** and c-IPN **h** cathodes.

chromatography using neutral aluminum oxide and the dichloromethane was removed by rotary evaporator. The obtained VAI-TFSI was dried overnight in a vacuum oven. The chemical structure of the VAI-TFSI was confirmed using $^1$H-NMR spectroscopy.

**Fabrication of c-IPN electrodes**

The electrode slurries were prepared with a composition ratio of LiNi$_{0.8}$Co$_{0.1}$Mn$_{0.1}$O$_2$(NCM811, LG energy solution)/carbon black additive (Super C65)/binder = 92/3/5 (w/w/w). For the control PVDF electrode, polyvinylidene fluoride (PVDF, Solvay) was solely used as binder and dissolved in N-methyl-2-pyrrolidone (NMP, Aldrich). For the control n-IPN and c-IPN electrodes, PVDF/trimethylolpropane triacrylate (TMPTA) (= 3/2 (w/w)) and PVDF/TMPTA/VAI-TFSI (= 3/0.7/1.3 (w/w/w)) were dissolved in NMP, respectively, in which benzyol peroxide (BPO, 1 wt%) was used as a thermal initiator. The electrodes were fabricated by casting the electrode slurry on an Al current collector. The casted electrode slurry was dried at 100 °C for 1 h and followed by roll-pressing at 90 °C. The electrode density of the electrodes examined herein was set at 2.8 g cc$^{-1}$ for a fair comparison. The electrodes were vacuum-dried at 120 °C for 12 h before the cell assembly.

**Structural characterization**

The electrochemical stability window of the crosslinked TMPTA and VAI-TFSI/TMPTA binders was investigated using linear sweep voltammetry (LSV) performed on a working electrode (stainless-steel) and a counter/reference electrode composed (Li-metal), and the LSV measurements were performed with a scan rate of 1.0 mV s$^{-1}$. The surface tension at the electrode slurry-gas interface was measured by a surface analyzer (Phoenix 300, SEO) using the pendant drop tensiometry. To ensure the reliability of the data from the zeta potential measurement (Zetasizer Nano ZS, Malvern), model suspensions were prepared at a dilute concentration (10 ppm) using NMP solvent. The

model suspensions were composed of a mixture of carbon conductive additive/binder = 3/5 (w/w) in NMP without including NCM811 powders, in which the composition ratio of the model suspensions was determined by considering the composition ratio (NCM811/carbon black additive/binder = 92/3/5 (w/w/w)) of the cathodes. The surface morphology and height deviation of the nonpressed electrodes were characterized using laser scanning confocal microscopy (FV1000, Carl Zeiss), in which the height deviation was the difference between an observed height of the electrode and its average height. The thermal crosslinking reaction of the c-IPN and n-IPN electrodes was examined using a Fourier transform infrared spectrometer (Alpha Platinum ATR, Bruker) with a spectral resolution of 4 cm$^{-1}$. The T$_g$ (glass transition temperature) of the binders were estimated by differential scanning calorimetry (Q2000, DuPont) at a heating rate of 20 °C min$^{-1}$. The tensile test of the binder films was conducted using a universal testing machine (DA-01, Petrol LAB) at a strain rate of 0.5 mm min$^{-1}$. The adhesion strength between the electrodes and Al current collectors was measured by a universal testing machine (DA-01, Petrol LAB) at a peel speed of 50 mm min$^{-1}$. For the electrolyte swelling measurement, the electrodes were immersed in the liquid electrolyte (1 M LiPF$_6$ in ethyl carbonate (EC)/diethyl carbonate (DEC) (=1/1 (v/v))) at 25 °C and their weight change was measured as a function of immersion time. The cross-sectional morphologies of the electrodes were characterized using a field emission scanning electron microscope (S-4800, Hitachi) in conjunction with an energy-dispersive X-ray spectrometer (JSM 6400, JEOL), in which the specimens were prepared by ion-milling system (IM4000, HITACHI). The internal structure of the pouch-type cell was characterized using X-ray microtomography (Xradia 520 Versa, Carl Zeiss Inc.). During tomography, 993 projection images were acquired by utilizing the × 4 optical magnification. To analyze the microstructure of the cycled NCM811 particles, cross-sectioned samples were thinned by using focused ion beam (Helios Nano Lab 450, FEI).

The structural and chemical analysis of the thinned samples was conducted using high resolution transmission electron microscopy (HR-TEM, ARM300, JEOL).

## Electrochemical and physicochemical characterizations

The electronic conductivity of the electrodes was measured using a four-point probe station (CMT-SR1000N, Advanced Instrument Tech). The electrochemical impedance spectroscopy (EIS) analysis of a symmetric cell (electrode‖electrode) at a fully lithiated state was conducted at a frequency range from $10^{-2}$ to $10^{6}$ Hz and an applied amplitude of 10 mV using potentiostat/galvanostat (VSP classic, Bio-Logic). The electrochemical performance of the electrodes was characterized using a 2032-type coin and pouch-type cells (NCM811 cathode‖polyethylene (PE) separator (thickness = 16 μm)‖Li-metal anode (thickness = 100 μm), liquid electrolyte (1 M LiPF$_6$ in EC/ DEC = 1/1 (v/v) with 10 wt.% fluoroethylene carbonate (FEC) and 1 wt.% vinylene carbonate (VC))) with electrolyte mass/cell capacity ($E/C$) ratio = 2.5 g Ah$^{-1}$, if not specified. The double-stacked pouch-type cell (composed of c-IPN cathode ($C/A$ of 18 mAh cm$^{-2}$)‖Li-metal anode ($C/A$ of 20 mAh cm$^{-2}$)) was fabricated using an Al pouch film as a packaging substance in a dry room with a dew point of −50 °C. The electrochemical performance of the pouch-type cell was evaluated at 25 °C under a fixed pressure set as 300 kPa. The specific energies and energy densities of the coin cells were estimated based on the weight/volume of cathode (including an Al current collector), anode (including a Cu current collector), separator, and electrolyte in the cell. Meanwhile, the specific energy and energy density of the pouch-type cell were estimated based on the experimentally measured weight/volume of the cell (including the pouch packaging film). Calculation details for these energy densities are described in Supplementary Tables 1 and 2. The galvanostatic intermittent titration technique (GITT) analysis was conducted at a current density of 0.1 C ( = 1.35 mA cm$^{-2}$) and with interruption time between each pulse of 30 min. The electrochemical reaction kinetics of the electrodes were investigated using cyclic voltammetry at a scan rate of 0.1 mV s$^{-1}$. The electrochemical performance cells were examined using a cycle tester (PEBC050.1, PNE Solution) at various charge/discharge conditions. The degree of delithiation of the electrodes in the through-thickness direction was confirmed using Raman spectroscopy (NRS-3100, JASCO). The oxidation state of the transition metal ions in the NCM811 particles was determined by estimating the intensity ratio of $E_g$ (the bending mode of metal-oxygen-metal in $a/b$-axis direction) to $A_{1g}$ (the stretching mode of metal-oxygen complex in $c$-axis direction). The structural evolution and electrostatic interaction between the electrodes and anions (or solvents) were traced by using Fourier transform infrared spectrometry (Alpha Platinum ATR, Bruker). The local environment of Li$^+$ in the electrodes was investigated using $^7$Li magic-angle spinning (MAS) nuclear magnetic resonance (NMR) spectroscopy (600 MHz FT-NMR, VNMRS 600, Agilent) with 1.6 mm HXY Fast MAS T3 probe and spinning at 20 kHz. The $^7$Li chemical shifts were referenced to a 1.0 M aqueous LiCl solution at $^7$Li (0 ppm) as an external standard. For the post-mortem analysis, the cells after 100 cycles were disassembled at the discharging state and rinsed with a solvent (DMC) to remove residual salts in an Ar-filled glovebox. The samples for the post-mortem analyses were transferred using the sealed pouch with inert gas. The chemcial change of the cycled NCM811 surface was analyzed using time-of-flight secondary ion mass-spectroscopy (TOF-SIMS 5, ION TOF) with Bi$_3^{2+}$ gun (25 keV, 1 pA) and X-ray photoelectron spectroscopy (ESCALAB 250XI, ThermoFisher) with focused monochromatized Al Kα radiation.

## Molecular dynamics simulation

The all-atom molecular dynamics (MD) simulations were performed using the Materials Studio 2019 R2 (BIOVIA Materials Studio) program. The COMPASSII force field[36] was used to describe interaction between LiPF$_6$, EC, DEC, VAI-TFSI, and TMTPA monomers. The van der Waals interactions were calculated using an atom-based cut-off method with cut-off of 12.5 Å, and the electrostatic interactions were treated with Ewald summation method[37]. The temperature and pressure were controlled by Nose-Hoover-Langevin (NHL) dynamics[38] and Berendsen barostat[39] and time coupling constants were adjusted to 1 and 0.1 ps, respectively. To investigate solvation characteristics of Li$^+$ at the vicinity of c-IPN and n-IPN binder units, modelling of the bulk polymer systems was conducted. 10 monomers in a chain are prepared to describe polymer binder chains and equilibrated. The detailed descriptions of each model system were summarized in Supplementary Table 6. The initial model systems were relaxed by steepest-descent geometry optimization. The convergence criteria for the optimization were set as 0.001 kcal mol$^{-1}$ for the maximum energy change and 0.5 kcal mol$^{-1}$ Å$^{-1}$ for the maximum force. After that, the canonical ensemble (i.e., NVT) simulation was performed for 500 ps at 298 K. In the second step, the isothermal-isobaric ensemble (i.e., NPT) simulation for 500 ps at 298 K, 1 bar were performed for equilibration. After equilibrating bulk structure of binder systems, we have expanded the cell and solvated each system with 1 M LiPF$_6$ in EC/DEC = 1/1 (v/v) to describe binder-electrolyte interface. Equilibration of solvated binder system was conducted in three steps. First, conducting geometry optimization as same as equilibration of bulk polymer system, second, the NVT simulation for 500 ps at 298 K, and finally NPT simulation for 4 ns at 298 K, 1 atm. Initial and final configurations for MD simulations are included in Supplementary Data 1–4. For the analysis of solvation characteristics, we used the radial distribution function (RDF, $g_{\alpha\beta}(r)$), which is shown as follows,

$$x_\alpha x_\beta \rho g_{\alpha\beta}(r) = \frac{1}{N}\left\langle \sum_{i=1}^{N_\alpha} \sum_{j=1}^{N_\beta} \delta(r - r_i - r_j) \right\rangle \quad (1)$$

where, $x_i$'s are the mole fractions of chemical type $i$, $N_i$'s are the number of atoms (or molecules) of chemical type $i$ or $j$, $N$ is the total number of atoms, and $\rho$ is the overall number density. A modified equation, which provides the coordination number ($n_{\alpha\beta}(r)$) derived from Eq. (2), is shown below,

$$n_{\alpha\beta}(r) = 4\pi\rho_\beta \int_{r_0}^{r_1} r^2 g_{\alpha\beta}(r) dr \quad (2)$$

where $\rho_\beta$ is the homogeneous number density of atom $\beta$ and the range of integrand between $r_0$ and $r_1$ represents the first solvation shell. All the analyses were performed for last 1 ns of NPT trajectories.

## Density functional theory calculation

To investigate highest occupied molecular orbital (HOMO) energy levels of Li$^+$ solvation structure at binder/electrolyte interfaces, density functional theory (DFT) calculations were performed using the DMol3 program[40,41]. The generalized gradient approximation with the Perdew-Burke-Ernzerhof (GGA-PBE) exchange-correlation functional[42] was used, and the core electrons were treated as all electrons with relativistic effects. To describe the van der Waals (vdW) interactions, Grimme's method[43] was applied for vdW energy correction. The spin polarization was incorporated in all calculations. The molecular orbitals were expanded by the double numerical plus polarization (DNP) 4.4 basis set with the global orbital cutoff of 5.1 Å. The convergence criterion for self-consistent calculation was set to $1 \times 10^{-6}$. The solution environment of the Li solvation shell was described using the conductor-like screening model (COSMO) method[44], applying the dielectric constant of ε = 13.637. Dielectric constant of solvation environment ($\varepsilon$) was calculated by following formula which describes

the dielectric constant of heterogeneous electrolyte mixture

$$\varepsilon = \left[ \left( \varepsilon_2^{1/3} - \varepsilon_1^{1/3} \right) v_2 + \varepsilon_1^{1/3} \right]^3 \qquad (3)$$

where $\varepsilon_i$'s are the dielectric constant of each electrolyte $i$, (i.e., 89.78 for ethyl carbonate and 2.81 for diethyl carbonate) and $v_2$ indicates the volume fraction of the corresponding electrolyte.

## Data availability

The authors declare that the data supporting the findings of this study are available within the paper, its Supplementary Information, and Supplementary Data files. Initial and final configurations for MD simulations are included in Supplementary Data 1–4. Extra data are available from the corresponding author upon request.

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

## Acknowledgements

This work was supported by the National Research Foundation of Korea (NRF) grant funded by the Korean Government (MSIT) (2021R1A2B5B03001615 and 2021R1A5A6002853). This work was also supported by the Technology Innovation Program (20010960) funded by the Ministry of Trade, Industry & Energy (MOTIE, Korea). Computational resources were supported by the National Supercomputing Center including technical support (KSC-2021-CRE-0539) and UNIST-HPC.

## Author contributions

J.-H.K. and S.-Y.L. designed this work. J.-H.K. performed the experimental characterization and electrochemical tests. J.W.K. synthesized cationic monomer. H.-S.M. participated in the Raman spectroscopy analysis. J.-H.K., K.M.L. and S.H.K. designed and performed theoretical calculations. T.Y., S.K.K. and S.-Y.L. supervised the overall project. All authors contributed to finalizing the manuscript.

## Competing interests

The authors declare no competing interests.
