## [Peer Review File · Nature Communications]

REVIEWER COMMENTS

Reviewer #1 (Remarks to the Author):

This work is of high significance to the lithium-ion battery (LIB) field, as thick electrodes are required to achieve major international industry and government cost reduction and energy density ultimate targets (i.e. >350 Wh/kg, <\$80/kWh, >2000 deep-discharge cycles, etc.). Most importantly, it addresses both of these goals simultaneously since thick electrodes reduce cost and raise cell energy density by eliminating a large fraction of the expensive current collector foils and separators.

It will be of interest LIB researchers, electrochemists, formulation chemists, and the energy storage community at large. The research results adequately support the conclusions of the binder functionality in LIB cathode environments, and the graphics in Figure 1a and 5g are highly illustrative. The methods are sound, and this data could easily be reproduced in other laboratories.

The added novelty of the binder properties in stabilizing the cathode dispersion, converting the dispersion to a stable coating network of particles with low agglomerate size, and providing an ionically conductive network in the dried electrode is particularly promising for the goal of preserving the rated gravimetric energy density of 376 Wh/kg at high discharge rates. Finally, the electrochemical performance data is of particular significance since it was paired with a real-world Li foil thickness of 100 microns. The binder seems to act almost like a polymer electrolyte with fixed PF6⁻ ions bound to the cationic polymer backbone, which then allow for Li⁺ hopping much in the same way protons transport in proton exchange membranes.

With that stated, it must be pointed out that the cycling performance data in Figures 1e and 6a were obtained at only 0.05C/-0.1C charge/discharge rates, which are not realistic operating conditions for some important applications. The higher observed areal capacity decay rate through only 50 cycles is likely due to the interaction of liquid carbonate electrolyte with anode Li metal and not the thick cathode itself. The authors might consider using a different, more stable anode in a future study to prove out the cathode advancements presented here where much higher cycling rates can be used that are in line with desired EV charging rates (2-6C) and discharging rates (0.5-2C). Please add discussion into the manuscript addressing this limitation.

The experimental methods used to characterize the effect of the cationic binder on both cathode dispersion stability and electrode structure, surface topology, electrode adhesion, and electrochemical performance were insightful and describe the added benefits well. However, more description of how the zeta potential magnitude was measured is relevant (i.e. Figure 2b), as the sample in NMP solvent was diluted to 10 ppm during the measurement which would also dilute the effect of the cationic binder

dispersant effect. What were the samples diluted with, and were the zeta potentials measured at different pH values?

Additional Comments:

In Introduction section, page 3, paragraph 2, it would be useful to add a reference to earlier Oak Ridge National Laboratory model [Z. Du et al., "Understanding limiting factors in thick electrode performance as applied to high energy density Li-ion batteries," *J Appl Electrochem* (2017) 47:405–415] that predicted the ionic transport limitations observed in the later references.

The right y-axis in Figures 1e and 6a should be changed to a 90-100% scale.

Reviewer #2 (Remarks to the Author):

The paper presents promising results regarding the performance of the binder for thick cathodes. The electrochemical performance of the thick cathode was extensively characterized through various spectroscopies and modeling efforts. I would like to provide some comments in hopes of clarifying the strength and weak points of the paper.

In Fig 4, the Raman spectra of the cross section of the thick cathode provides compelling evidence of an inhomogeneous state of charge (SOC) for the n-IPN cathode. However, it is unclear what the target SOC is for the entire cathode. For instance, it is unknown whether the data was collected after the cathodes were completely discharged.

Figure 1 provides clear evidence that potential degradation of both the Li metal and separator could be the root cause of capacity fading. However, I am unsure if the liquid electrolyte was replaced during the re-fabrication at the 50th cycle. As the authors have already analyzed the electrochemical data, it would be beneficial to compare the dQ/dV profiles, voltage profiles, and EIS to elaborate on the discussion regarding the root cause of capacity fading. One possibility could be electrolyte consumption due to repeated Li SEI formation, leading to a significant increase in cell impedance, which may result in incomplete charging of the cell at the given cut-off voltage.

The improvement mechanism of c-IPN binder was proposed as follows;

"These results demonstrate that the c-IPN binder, driven by its electrostatic attraction with PF6 and electrostatic repulsion with Li⁺, enabled the prevalence of freely mobile Li⁺ and uniform distribution of

Li⁺ concentration." This hypothesis was supported by RDF and MAS NMR data. Based on the proposed mechanism, I have identified a logical flaw that could result in self-contradiction. The binder, which serves as electrostatic repulsion for Li ions, can hinder the charge transfer process and the transport of Li ions by being present on the surface of the cathode active material. Additionally, there is not a direct correlation between the repulsion of Li ions and their long-range transport behavior through liquid channels in the cathode pores, which is a logical leap.

To better comprehend the Li-ion transfer and improve Li transport, it is crucial to have a clear understanding of the microstructure of thick cathodes. For instance, the porosity of the electrodes plays a crucial role in determining the performance of the battery. Unfortunately, there is no mention of either the porosity or areal density in the current description, and it would be beneficial to include this information.

Another improvement mechanism proposed was proposed as follows; "Consequently, PF6, which is adjacent to the c-IPN binder, was vulnerable to decomposition, thus promoting the formation of PF6-derived LiF-rich CEI layers." LiF enriched CEI layer may not positively impact the conductivity of CEI due to its low Li-ion conductivity. However, electrochemical data did not support this phenomenon and EIS fitting did not consider charge transfer impedance at all (SI Fig. 17).

Overall, I observed that there are scattered assumptions regarding the improvement mechanism, which are not cohesively linked to each other. Additionally, the proposed mechanisms lack strong support from the electrochemical data.

REVIEWER COMMENTS

Reviewer #1 (Remarks to the Author):

This work is of high significance to the lithium-ion battery (LIB) field, as thick electrodes are required to achieve major international industry and government cost reduction and energy density ultimate targets (i.e. >350 Wh/kg, <\$80/kWh, >2000 deep-discharge cycles, etc.). Most importantly, it addresses both of these goals simultaneously since thick electrodes reduce cost and raise cell energy density by eliminating a large fraction of the expensive current collector foils and separators.

It will be of interest LIB researchers, electrochemists, formulation chemists, and the energy storage community at large. The research results adequately support the conclusions of the binder functionality in LIB cathode environments, and the graphics in Figure 1a and 5g are highly illustrative. The methods are sound, and this data could easily be reproduced in other laboratories.

The added novelty of the binder properties in stabilizing the cathode dispersion, converting the dispersion to a stable coating network of particles with low agglomerate size, and providing an ionically conductive network in the dried electrode is particularly promising for the goal of preserving the rated gravimetric energy density of 376 Wh/kg at high discharge rates. Finally, the electrochemical performance data is of particular significance since it was paired with a real-world Li foil thickness of 100 microns. The binder seems to act almost like a polymer electrolyte with fixed PF6⁻ ions bound to the cationic polymer backbone, which then allow for Li⁺ hopping much in the same way protons transport in proton exchange membranes.

With that stated, it must be pointed out that the cycling performance data in Figures 1e and 6a were obtained at only 0.05C/-0.1C charge/discharge rates, which are not realistic operating conditions for some important applications. The higher observed areal capacity decay rate through only 50 cycles is likely due to the interaction of liquid carbonate electrolyte with anode Li metal and not the thick cathode itself. The authors might consider using a different, more

stable anode in a future study to prove out the cathode advancements presented here where much higher cycling rates can be used that are in line with desired EV charging rates (2-6C) and discharging rates (0.5-2C). Please add discussion into the manuscript addressing this limitation.

→ Thank you so much for the reviewer's insightful comments. As the reviewer pointed out, a more stable anode with electrochemical reliability should be considered in future studies to prove out the cathode advancements for practical battery applications that require longer cycling retention and faster current rates. As the reviewer is aware of, it is difficult to find out electrochemically stable anodes that can be combined with high-areal-capacity cathodes (e.g., 20 mAh cm⁻² of this study). Potential approaches implemented to address the issue of anodes include development of advanced Li-metal anodes (including the Li hosts, protective layers, alloys, and others) and advanced Si anodes with suppressed volume expansion. In response to the reviewer's comment, the issue associated with the anodes was added in the revised manuscript.

[Revised manuscript]

“Moreover, the cell exhibited a stable cycling retention without the use of unconventional electrolytes that are specially designed for Li-metal anodes. Particularly, the cell almost completely recovered its initial C/A after the replacement of the cycled Li-metal anode, liquid electrolyte, and separator with fresh ones (Fig. 1e and Supplementary Fig. 2), indicating that the c-IPN cathode is not the major cause of the cycling decay. A supplementary experiment was conducted to further identify a major cause of this decline in the cycling retention (Supplementary Fig. 4). Replacing the cycled liquid electrolyte with a fresh one, while leaving the cycled Li-metal anode unchanged, failed to return to the initial voltage profile and showed rapid capacity fading with cycling. In comparison, replacing the cycled Li-metal anode with a fresh one returned to the normal and stable voltage profile. This result exhibits that the cycled Li-metal anode has a critical effect on the cycling degradation of the Li-metal full cell with the high-C/A cathode. Thus, coupling the high-C/A cathodes with electrochemically stable high-capacity anodes (such as advanced Li-metal or Si) should be conducted in future studies to highlight the cathode advancements for practical battery applications that require longer cycling retention and faster current rates.”

The experimental methods used to characterize the effect of the cationic binder on both cathode dispersion stability and electrode structure, surface topology, electrode adhesion, and electrochemical performance were insightful and describe the added benefits well. However, more description of how the zeta potential magnitude was measured is relevant (i.e. Figure 2b), as the sample in NMP solvent was diluted to 10 ppm during the measurement which would also dilute the effect of the cationic binder dispersant effect. What were the samples diluted with, and were the zeta potentials measured at different pH values?

→ We appreciate the reviewer's valuable comments. As the reviewer is aware of, high concentrations or large particle size of samples for the zeta potential measurement often lead to unwanted data fluctuation, so dilution of the samples is needed to obtain data accuracy and reliability (ref: S. Bhattacharjee et al., *J. Control. Release* **235** 337 (2016)).

Both DLS and ZP measurements are based on light scattering and hence, only clear samples can be subjected to these two techniques. Additionally, both these techniques are not capable to handle concentrated samples. Just to exemplify, the *Stokes-Einstein equation* – which is the backbone of particle size measurements based on light scattering – is only mathematically feasible at infinitely dilute concentrations. In reality usually 50–100 µg/ml concentrations are used. Unfortunately, it

Fig. A captured image from a review article published in *J. Control. Release* **235**, 337 (2016), “DLS and zeta potential – What they are and what they are not?”. 1 µg mL corresponds to 1 ppm.

To this end, model suspensions were prepared at a dilute concentration (10 ppm, as described in the experimental section) using NMP solvent. The model suspensions were composed of a mixture of carbon conductive additive/binder = 3/5 (w/w) in NMP without the inclusion of NCM811 powders, in which the composition ratio of the model suspensions was determined by considering the composition ratio (NCM811/carbon black additive/binder = 92/3/5 (w/w/w))

of the cathodes. The NCM811 powders were excluded due to their large particle size ($\sim 10 \mu\text{m}$), which could negatively affect the reliability of the data as described above.

In response to the reviewer's comment, the details of the zeta potential measurement and model suspension preparation were described in the revised manuscript. In addition, to address the reviewer's comment on the dilution, we prepared other model suspensions with a higher concentration (10 \rightarrow 100 ppm). The zeta potential of the high-concentration suspensions appeared to be consistent with the result of the low-concentration suspensions, verifying the presence of the cationic moiety in the c-IPN binder. Please see the figure below.

Fig. Absolute zeta potential values of the model suspensions with a concentration of 100 ppm.

In response to the reviewer's comment, we attempted to measure the zeta potential values of the model suspensions at different pH values. However, we found that the PVDF, which is a component of the n-IPN and c-IPN, was not dissolved in water, which thus made it difficult for us to prepare the samples for this measurement. The reviewer's kind understanding is highly appreciated, in advance.

[Revised manuscript]

“The reduction of the surface tension of electrode slurries is known to alleviate drying-triggered internal stress which causes electrode cracking problem¹⁷. The c-IPN electrode slurry exhibited lower surface tension compared to the control electrode slurries (**Fig. 2a** and **Supplementary**

Fig. 9), indicating that the c-IPN binder acted as an ionic surfactant. The presence of the cationic moiety in the c-IPN binder was identified by conducting zeta potential analysis. For this measurement, model suspensions (carbon conductive additive/binder = 3/5 (w/w) in NMP solvent) were prepared at a low concentration of 10 ppm, details of which are described in the experimental. The model suspension with the c-IPN binder showed a higher absolute zeta potential value (~ 14 mV) compared to the control suspensions with the neutral binders (~ 1 mV) (Fig. 2b), demonstrating the cationic nature of the c-IPN binder. It is known that the high absolute zeta potential value of a suspension solution is beneficial for improving the dispersion state¹⁸.”

“Structural characterization

The electrochemical stability window of the crosslinked TMPTA and VAI-TFSI/TMPTA binders was investigated using linear sweep voltammetry (LSV) performed on a working electrode (stainless-steel) and a counter/reference electrode composed (Li-metal), and the LSV measurements were performed with a scan rate of 1.0 mV s⁻¹. The surface tension at the electrode slurry-gas interface was measured by a surface analyzer (Phoenix 300, SEO) using the pendant drop tensiometry. To ensure the reliability of the data from the zeta potential measurement (Zetasizer Nano ZS, Malvern), model suspensions were prepared at a dilute concentration (10 ppm) using NMP solvent. The model suspensions were composed of a mixture of carbon conductive additive/binder = 3/5 (w/w) in NMP without including NCM811 powders, in which the composition ratio of the model suspensions was determined by considering the composition ratio (NCM811/carbon black additive/binder = 92/3/5 (w/w/w)) of the cathodes.”

Additional Comments:

In Introduction section, page 3, paragraph 2, it would be useful to add a reference to earlier

Oak Ridge National Laboratory model [Z. Du et al., "Understanding limiting factors in thick electrode performance as applied to high energy density Li-ion batteries," *J Appl Electrochem* (2017) 47:405–415] that predicted the ionic transport limitations observed in the later references.

→ Thank you for the reviewer's valuable comment. The paper mentioned by the reviewer reported the kinetic limiting factor of thick electrodes by numerical modelling. This numerical study showed that Li⁺ depletion in the electrode is more pronounced for thicker electrodes, which is well matched with our experimental results. In response to the reviewer's comment, this paper was added in the reference of the revised the manuscript.

[Revised manuscript]

“To achieve high-C/A electrodes (= areal-mass-loading (M/A) × specific capacity of electrode active materials (C_{sp}))⁴, the M/A should be maximized while stably maintaining the C_{sp}. However, owing to the use of thick electrodes (physical issue) and non-uniform charge transfer throughout the electrodes (electrochemical issue), conventional electrodes cannot achieve this requirement⁵. Particularly, the drying of processing solvents, such as N-methyl pyrrolidone (NMP) and water, during the fabrication of thick electrodes often induces crack formation and delamination from metallic current collectors, thus limiting the increase in the M/A values^{6,7}. Additionally, with an increase in the electrode thickness, charge transfer in electrode active materials tends to demonstrate uneven and sluggish reaction kinetics in the through-thickness direction of the electrodes, resulting in the loss of the C_{sp} values⁸⁻¹⁰.”

“References

- 8 Z. Du. et al. Understanding limiting factors in thick electrode performance as applied to high energy density Li-ion batteries. *J. Appl. Electrochem.* **47**, 405 (2017).
- 9 Heubner, C., Schneider, M. & Michaelis, A. Diffusion-Limited C-Rate: A Fundamental Principle Quantifying the Intrinsic Limits of Li-Ion Batteries. *Adv. Energy Mater.* **10** (2019).
- 10 Park, K.-Y. et al. Understanding capacity fading mechanism of thick electrodes for

lithium-ion rechargeable batteries. *J. Power Sources* **468**, 228369 (2020).”

The right y-axis in Figures 1e and 6a should be changed to a 90-100% scale.

→ Many thanks for the reviewer’s valuable comment. In response to the reviewer’s comment, the scale of the right y-axis (coulombic efficiency) was changed.

[Revised manuscript]

Fig. 1 | e, Cycling performance of the double-stacked pouch-type cell at charge/discharge current density of 0.9 mA cm⁻²/1.8 mA cm⁻² and voltage range of 3.0–4.4 V.

Fig. 6 | a, Cycling performance of the Li-metal full cells (n-IPN cathode vs. c-IPN cathode), in which the cathodes with an M/A of 65.0 mg cm^{-2} (corresponding to $C/A = 13.5 \text{ mAh cm}^{-2}$) were assembled with Li-metal anodes ($C/A = 20.0 \text{ mAh cm}^{-2}$, N/P ratio = 1.5). The cells were cycled at a charge/discharge current density of $0.68/1.35 \text{ mA cm}^{-2}$ and voltage range of 3.0–4.4 V.

Reviewer #2 (Remarks to the Author):

The paper presents promising results regarding the performance of the binder for thick cathodes. The electrochemical performance of the thick cathode was extensively characterized through various spectroscopies and modeling efforts. I would like to provide some comments in hopes of clarifying the strength and weak points of the paper.

In Fig 4, the Raman spectra of the cross section of the thick cathode provides compelling evidence of an inhomogeneous state of charge (SOC) for the n-IPN cathode. However, it is unclear what the target SOC is for the entire cathode. For instance, it is unknown whether the data was collected after the cathodes were completely discharged.

→ Thank you for the reviewer's valuable comments. In this study, the cells were charged to 4.4 V (corresponding to 100% SOC) at a current density of 1.1 mA cm^{-2} , and then disassembled to collect the charged cathodes, according to the relevant references (ref.: J. Cho et al., *Adv. Mater.* **32**, 2003040 (2020) and E. J. Berg et al., *Front. Energy Res.* **6**, 82 (2018)). The difference in the local SOC values of the cathodes in their through-thickness direction was investigated by measuring the intensity ratio of E_g/A_{1g} in the Raman spectra. In response to the reviewer's comment, the experimental details for the Raman spectroscopy were added in the revised manuscript.

Figure S7. Raman spectra of PC-NCM during the extraction of lithium.

cycling (Figure S7, Supporting Information). Aside from the noticeable change in the spectrum shape, it should be emphasized that the E_g/A_{1g} intensity ratio of the PC-NCM electrode clearly showed SOC heterogeneity according to the position in the electrode (Figure 2e). The value increases as the detection point in the electrode increases from the bottom to the top region. We inferred that larger amounts of lithium ions were extracted at the electrode surface than those inside the electrode during cycling. This implies that the high lithium ion flux near the electrode surface could induce the electrode potential to reach the charge cut-off voltage earlier than the other side within the electrode. This relatively high potential environ-

Fig. A captured image from a research article published in J. Cho et al., *Adv. Mater.* **32**, 2003040 (2020), “Boosting Reaction Homogeneity in High-Energy Lithium-Ion Battery Cathode Materials”. Raman spectroscopy of the NCM active materials as a function of delithiation (SOC).

[Revised manuscript]

Fig. 4 | **d**, Cross-sectional optical micrograph of the charged c-IPN cathode ($M/A = 65 \text{ mg cm}^{-2}$), in which the cells were charged to 4.4 V at a current density of 1.1 mA cm^{-2} , and then disassembled to collect the charged cathode. The two boxes (indicating the top and bottom regions) in the image were selected for the Raman spectroscopy. **e,f**, Raman spectra and intensity ratio of E_g/A_{1g} of the n-IPN (**e**) and c-IPN (**f**) cathodes with an M/A of 65 mg cm^{-2} at the top and bottom regions, in which E_g and A_{1g} represent the bending mode of metal-oxygen-metal in a/b -axis direction and stretching mode of metal-oxygen complex in c -axis direction, respectively. For this analysis, the cells were charged to 4.4 V (corresponding to 100% SOC) at a current density of 1.1 mA cm^{-2} , and then disassembled to collect the charged cathode.

“To further elucidate the difference in the C_{sp} of the c-IPN and control n-IPN cathodes, their state of charge (SOC) was analyzed in the through-thickness direction. The local SOC values of the cathodes were examined as a function of the distance from the Al current collectors (denoted as “top” and “bottom”) using Raman spectroscopy (**Fig. 4d** and **Supplementary Fig. 15**). It is known that the intensity ratio of E_g/A_{1g} in the Raman spectra tends to increase with increasing SOC (i.e., degree of de-lithiation) of cathode active materials²⁵. At 100% SOC, the control n-IPN cathode exhibited a lower E_g/A_{1g} intensity ratio in the bottom region (**Fig. 4e**), indicating that the NCM811 does not easily undergo de-lithiation in the bottom region (adjacent to the Al current collector) owing to the uneven and tortuous pathway of Li^+ in the cathode. In contrast, the E_g/A_{1g} intensity ratio of the c-IPN cathode remained almost unchanged regardless of its cross-sectional regions, verifying the through-thickness uniformity in the redox reaction of NCM811 (**Fig. 4f**). This redox uniformity of the c-IPN cathode was attributed to the facile Li^+ migration (rather than the electron conduction) in the through-thickness direction (**Supplementary Fig. 16** and **17**).”

Figure 1 provides clear evidence that potential degradation of both the Li metal and separator could be the root cause of capacity fading. However, I am unsure if the liquid electrolyte was replaced during the re-fabrication at the 50th cycle. As the authors have already analyzed the electrochemical data, it would be beneficial to compare the dQ/dV profiles, voltage profiles, and EIS to elaborate on the discussion regarding the root cause of capacity fading. One possibility could be electrolyte consumption due to repeated Li SEI formation, leading to a significant increase in cell impedance, which may result in incomplete charging of the cell at the given cut-off voltage.

→ We are grateful for the reviewer's insightful comments. In **Fig. 1e**, in addition to the replacement of the cycled Li-metal anode and separator, the cycled liquid electrolyte was also replaced with a fresh one. A similar experiment for the replacement of the cycled Li-metal

anode, separator, and liquid electrolyte was reported in previous studies for Li-metal cells (ref.: A. Manthiram et al., *Adv. Energy Mater.* **6**, 1502459 (2016), and S. Y. Lee et al., *Nat. Commun.* **13**, 2541 (2022)). In response to the reviewer's comment, an additional description on the replacement of the cycled cell components was provided in the revised manuscript.

In response to the reviewer's comment, the voltage and dQ/dV profiles of the double-stacked pouch-type Li-metal full cells were examined as a function of cycle number. As the cycle number was increased (see **Supplementary Fig. 2a** of the revised manuscript), the cell tended to quickly reach a cut-off voltage together with the increased overpotential. This result was verified by the sluggish redox kinetics in the dQ/dV profiles (see **Supplementary Fig. 2b** of the revised manuscript). Meanwhile, when the cycled cathode was reassembled with the fresh Li-metal anode, liquid electrolyte, and separator, the resulting Li-metal full cell returned to normal charge/discharge voltage and dQ/dV profiles (see **Supplementary Fig. 2**).

In response to the reviewer's comment, we identified the root cause of the capacity fading during the charge/discharge cycling (see **Supplementary Fig. 4**). To this end, three model cells were prepared as follows: model cell #1 (cycled cathode||fresh Li-metal anode in fresh liquid electrolyte) vs. model cell #2 (cycled cathode||fresh Li-metal anode in cycled liquid electrolyte) vs. model cell #3 (cycled cathode||cycled Li-metal anode in fresh liquid electrolyte). The model cell #1 and #2 returned to the initial voltage profile together with the high C/A value ($\sim 18 \text{ mAh cm}^{-2}$). In comparison, the model cell #3 failed to return to the initial voltage profile even in the presence of a fresh liquid electrolyte and showed rapid capacity fading with cycling. This result demonstrates that the cycled Li-metal anode is a major cause for the cycling degradation of the Li-metal full cell with the high-C/A cathode. In response to the reviewer's comment, this new result on the root cause of the capacity degradation in **Fig. 1e** was added in the revised manuscript. The reviewer's valuable comment is highly appreciated, again.

[Revised manuscript]

Fig. 1 | e, Cycling performance of the double-stacked pouch-type cell at charge/discharge current density of $0.9 \text{ mA cm}^{-2}/1.8 \text{ mA cm}^{-2}$ and voltage range of 3.0–4.4 V.

Supplementary Fig. 2 | a,b, Galvanostatic charge/discharge profiles (a) and corresponding dQ/dV plots (b) of the double-stacked pouch cell at 2nd, 40th, and 50th (after reassembly) cycles, in which the cell was cycled at charge/discharge current density of $0.9 \text{ mA cm}^{-2}/1.8 \text{ mA cm}^{-2}$ and voltage range of 3.0–4.4 V. The cycled cathodes were reassembled with fresh Li-metal anode, separator, and liquid electrolyte at 50th cycle.

Supplementary Fig. 4 | a-c. Galvanostatic charge/discharge profiles of the reassembled pouch cells: A model cell #1 (cycled cathode||fresh Li-metal anode in fresh liquid electrolyte) (a), A model cell #2 (cycled cathode||fresh Li-metal anode in cycled liquid electrolyte) (b), and a model cell #3 (cycled cathode||cycled Li-metal anode in fresh liquid electrolyte) (c). The cells were cycled at charge/discharge current rate of 0.05 C/0.1 C (= 0.9 mA cm⁻²/1.8 mA cm⁻²) and voltage range of 3.0 – 4.4 V.

To identify a major cause of the cycling fading of the Li-metal full cells shown in Fig. 1e, three model cells were prepared as follows: model cell #1 (cycled cathode||fresh Li-metal anode in fresh liquid electrolyte) vs. model cell #2 (cycled cathode||fresh Li-metal anode in cycled liquid electrolyte) vs. model cell #3 (cycled cathode||cycled Li-metal anode in fresh liquid electrolyte). The model cell #1 and #2 returned to an initial voltage profile together with the high C/A value (~ 18 mAh cm⁻²). In comparison, the model cell #3 failed to return to the initial voltage profile even in the presence of a fresh liquid electrolyte and showed rapid capacity fading with cycling. This result exhibits that the cycled Li-metal anode is a major cause of the cycling degradation of the full cells with the high-C/A cathodes.

[Revised manuscript]

“Moreover, the cell exhibited a stable cycling retention without the use of unconventional electrolytes that are specially designed for Li-metal anodes. Particularly, the cell almost completely recovered its initial C/A after the replacement of the cycled Li-metal anode, liquid electrolyte, and separator with fresh ones (Fig. 1e and Supplementary Fig. 2), indicating that

the c-IPN cathode is not the major cause of the cycling decay. A supplementary experiment was conducted to further identify a major cause of this decline in the cycling retention (Supplementary Fig. 4). Replacing the cycled liquid electrolyte with a fresh one, while leaving the cycled Li-metal anode unchanged, failed to return to the initial voltage profile and showed rapid capacity fading with cycling. In comparison, replacing the cycled Li-metal anode with a fresh one returned to the normal and stable voltage profile. This result exhibits that the cycled Li-metal anode has a critical effect on the cycling degradation of the Li-metal full cell with the high-C/A cathode. Thus, coupling the high-C/A cathodes with electrochemically stable high-capacity anodes (such as advanced Li-metal or Si) should be conducted in future studies to highlight the cathode advancements for practical battery applications that require longer cycling retention and faster current rates.”

The improvement mechanism of c-IPN binder was proposed as follows;

"These results demonstrate that the c-IPN binder, driven by its electrostatic attraction with PF6 and electrostatic repulsion with Li⁺, enabled the prevalence of freely mobile Li⁺ and uniform distribution of Li⁺ concentration." This hypothesis was supported by RDF and MAS NMR data. Based on the proposed mechanism, I have identified a logical flaw that could result in self-contradiction. The binder, which serves as electrostatic repulsion for Li ions, can hinder the charge transfer process and the transport of Li ions by being present on the surface of the cathode active material. Additionally, there is not a direct correlation between the repulsion of Li ions and their long-range transport behavior through liquid channels in the cathode pores, which is a logical leap.

→ Many thanks for the reviewer’s valuable comments on the improvement mechanism of the c-IPN binder. If a cationic binder is allowed to completely cover the entire surface of cathode active materials as a kind of wrapping layer, the binder may hinder Li⁺ access to the cathode active materials and the charge transfer process, as the reviewer pointed out. However, in real electrodes, binders are present as an adhesive between the cathode active materials, while a

large part of the cathode active material remains uncovered by the binder, which thus allows for the participation in the electrochemical reaction when exposed to liquid electrolytes (ref.: J. W. Choi et al., *Nat. Rev. Mater.* **1**, 1 (2016), J. W. Choi et al., *J. Mater. Chem. A* **1**, 15224 (2013)). A similar morphology was observed in the c-IPN electrode. Please see a SEM image below.

Fig. A captured image from a review article published in *Nat. Rev. Mater.* **1**, 1 (2016) “Promise and reality of post-lithium-ion batteries with high energy densities”. The schematic shows the morphology of the polymer binder and the intermolecular interactions.

Fig. A captured image from a research article published in *J. Mater. Chem. A* **1**, 15224 (2013) “Improved cycle lives of LiMn_2O_4 cathodes in lithium ion batteries by an alginate biopolymer from seaweed”.

Fig. A captured image from a research article published in *ACS Appl. Mater. Interfaces* **14**, 16245 (2022) “One Stone for Multiple Birds: A Versatile Cross-Linked Poly(dimethyl siloxane) Binder Boosts Cycling Life and Rate Capability of an NCM 523 Cathode at 4.6 V”. The schematic shows the morphology of the polymeric binder in the electrode.

Fig. A SEM image showing the binder morphology of the c-IPN electrode in this study, in which the yellow dotted regions indicate the presence of the binder. A significant part of the NCM811 powders remains uncovered by the binder, which can thus participate in the electrochemical reaction when exposed to liquid electrolytes.

Based on the understanding of the binder morphology in the electrode described above, the original manuscript discussed the effect of the binder on the Li^+ transport through liquid electrolyte-filled interstitial pores formed between the NCM811 powders. In response to the reviewer's comment, additional explanation was provided in the revised manuscript to clearly demonstrate the effect of the c-IPN binder.

[Revised manuscript]

“Binders are known to act as an adhesive between the cathode active materials (not completely covering them), while leaving a large portion of the cathode active material exposed to electrolytes²⁶. Thus, the c-IPN binder could affect the surface charge environment in the electrolyte-filled interstitial pores of the c-IPN electrode, thereby contributing to the facile Li^+ migration described above. The imidazolium-based cationic groups of the c-IPN binder enabled electrostatic attraction^{19,27} with the anion (PF_6^-) of liquid electrolyte (1 M LiPF_6 in ethyl carbonate (EC)/diethyl carbonate (DEC) = 1/1 (v/v)).”

In response to the reviewer's comment, additional investigations on the Li^+ transport in the electrolyte-filled pores of the cathodes were conducted.

1) Inversion-recovery plots obtained from ^7Li MAS NMR spectra

: We investigated the Li^+ mobility in the electrolyte-filled pores of the cathodes by analyzing inversion-recovery plots of ^7Li MAS NMR spectra. From the normalized intensities of ^7Li -NMR spectra (I/I_0) plotted as a function of time, spin-lattice relaxation time (T_1) values were obtained (**Supplementary Fig. 18**). A smaller T_1 value is known to reflect a faster diffusion rate of ions. The c-IPN cathode showed a smaller T_1 value (737 ms) than the n-IPN cathode (957 ms), indicating the faster Li^+ mobility in the electrolyte-filled pores. This result was added in the revised manuscript to verify the beneficial effect of the c-IPN binder on Li^+ transport phenomena in the electrolyte-filled pores of the cathodes.

[Revised manuscript]

Supplementary Fig. 18 | Inversion-recovery plots obtained from ^7Li MAS NMR spectra. The spin-lattice relaxation time (T_1) was calculated using the following equation¹⁰:

$$I = I_0(1 - \exp^{-t/T_1})$$

where I is the peak intensity at time t , I_0 is the saturation intensity, respectively.

“The local environment of Li^+ in electrolyte-filled pores of the c-IPN cathode was identified using ^7Li magic-angle spinning (MAS) nuclear magnetic resonance (NMR) spectroscopy (**Fig. 5b**). Compared to the control n-IPN cathode, the c-IPN cathode exhibited a downfield shift and a narrower width in the ^7Li spectrum, indicating the improvement in the dissociation of Li salts²⁸ and the mobility of free Li^{+29} , respectively. From the normalized intensities of ^7Li -NMR spectra (I/I_0) plotted as a function of time, spin-lattice relaxation time (T_1) values were obtained (**Supplementary Fig. 18**). A smaller T_1 value is known to reflect faster diffusion rate of ions³⁰. The c-IPN cathode showed a smaller T_1 value (737 ms) than the n-IPN cathode (957 ms), verifying the faster Li^+ mobility in the electrolyte-filled pores.”

2) In-depth discussion on the EIS analysis of the symmetric cell (electrode|separator|electrode) : Nyquist plots of symmetric cells at 0% SOC and the corresponding curves fitted by a transmission line equivalent circuit model (TLM) are known to describe non-faradaic processes in the electrolyte-filled porous electrodes. In these Nyquist plots, the projection of a linear slope to the real axis is defined as the ionic resistance in the electrolyte-filled pores of electrodes

$(R_{ion}/3)$ (ref.: Y. Ukyo et al., *J. Electrochem. Soc.* **159**, A1034 (2012), Y. Takeuchi et al., *J. Phys. Chem. C* **119**, 4612 (2015), and X. Duan et al., *Science* **356**, 599 (2017)).

Figure 1. Schematic representations of electrode structures and their equivalent circuit models.

Fig. A captured image from a research article published in *J. Phys. Chem. C* **119**, 4612 (2015), “Impedance Spectroscopy Characterization of Porous Electrodes under Different Electrode Thickness Using a Symmetric Cell for High-Performance Lithium-Ion Batteries”.

Figure 3. Simulated Nyquist plots for a cylindrical pore, $L = 1$ cm, $r = 0.5$ cm, $C_{dl} = 0.1$ mF cm $^{-2}$, as predicted by: (a) Eq. (1), $R_{ion,L} = 100$ Ω cm $^{-1}$, (b) Eq. (4), $R_{ion,L} = 100$ Ω cm $^{-1}$, $R_{ct,A} = 300$ Ω cm 2 , (c) Eq. (4), $R_{ion,L} = 100$ Ω cm $^{-1}$, $R_{ct,A} = 3000$ Ω cm 2 . Frequency range: 100 kHz–100 mHz.

Fig. A captured image from a research article published in *J. Electrochem. Soc.* **159**, A1034 (2012), “Theoretical and Experimental Analysis of Porous Electrodes for Lithium-Ion Batteries

by Electrochemical Impedance Spectroscopy Using a Symmetric Cell”.

Fig. S7. Nyquist plots for composites with tunable in-plane nanopores by using a symmetric cell with two identical electrodes (11 mg cm^{-2}) for a state of charge (SOC) at 0% (hollow symbols) (A-D). The solid lines are the best-fitting simulations for the equivalent circuits using the generalized finite length Warburg element open circuit terminus (W_o) as shown in (E) (36). W_o is used to simulate $R_{ion}/3$, and its fitting parameter R_{W_o} corresponds to the value of $R_{ion}/3$. The different impedances determined as $R_{ion}/3$, R_{high} and R_{sol} for the various electrodes are listed in the table S1. The inset in (B) shows how $R_{ion}/3$ is determined.

Fig. A captured image from a research article published in *Science* **356**, 599 (2017), “Three-dimensional holey-graphene/niobia composite architectures for ultrahigh-rate energy storage”.

Meanwhile, it is known that high C/A electrodes tend to provide longer Li^+ migration paths than relatively low C/A electrodes. Thus, for high C/A electrodes, Li^+ transport in the electrolyte-filled pores of the electrodes, which is represented by R_{ion} , could have a greater effect on the cell performance than charge transfer (represented by R_{ct}). Please see the figure below (ref.: Takeuchi and coworkers *J. Phys. Chem. C* **119**, 4612 (2015)).

Fig. A captured image from a research article published in *J. Phys. Chem. C* **119**, 4612 (2015), “Impedance Spectroscopy Characterization of Porous Electrodes under Different Electrode Thickness Using a Symmetric Cell for High-Performance Lithium-Ion Batteries”. By analyzing the impedance of ion transport through the electrolyte-filled pores of the electrodes and charge transfer as a function of electrode thickness (i.e., areal-mass loading of the electrode active material), it is shown that the electrochemical performance of thick electrodes is significantly influenced by ion transport in the electrolyte-filled pores of the electrodes rather than by charge transfer.

Based on the above-mentioned understanding, in **Fig. 5e**, the ion conductivity ($\sigma_{\text{electrode}}$) in the electrolyte-filled pores of the cathodes was calculated by analyzing the Nyquist plot of the symmetric cells. The c-IPN cathode showed a higher $\sigma_{\text{electrode}}$ over a wide range of temperatures and a lower activation energy (E_a) compared to the control n-IPN cathode. An additional explanation on the ion conduction in the electrolyte-filled pores of the cathodes and details of the EIS analysis of the symmetric cell were provided in the revised manuscript.

[Revised manuscript]

“To examine the ion conductivity ($\sigma_{\text{electrode}}$) **in the electrolyte-filled pores of electrodes**, the electrochemical impedance spectroscopy (EIS) analysis of the symmetric cells was performed (**Supplementary Fig. 19**). **A slope observed in the low frequency region of Nyquist plots is known to reflect ionic resistance in the electrolyte-filled pores of electrodes ($R_{\text{ion}}/3$, $\sigma_{\text{electrode}} = 1/R_{\text{ion}}$)³¹.** The c-IPN cathode exhibited a higher $\sigma_{\text{electrode}}$ over a wide range of temperatures and

a lower activation energy (E_a) compared to the control n-IPN cathode (**Fig. 5e**).”

Fig. 5 | e, Arrhenius plots for the ionic conductivities **in the electrolyte-filled pores of** the c-IPN (vs. control n-IPN) cathodes ($M/A = 65 \text{ mg cm}^{-2}$), which were obtained by the electrochemical impedance spectroscopy (EIS) analysis of the symmetric cells (electrode|separator|electrode) at 0% SOC.

3) GITT analysis of the Li-metal full cells (cathode|separator|Li-metal anode)

: The result of the ion conductivity ($\sigma_{\text{electrode}}$) in the electrolyte-filled pores of the cathode was further verified by analyzing the GITT profiles of the cells (**Fig. 5f** and **Supplementary Fig. 20**) and Li^+ diffusion coefficient (D_{Li^+}) (**Supplementary Table 4**). The cell with the c-IPN cathode exhibited lower internal resistances (R_{internal}) during discharge reactions and higher Li^+ diffusion coefficients (D_{Li^+}) compared to the control n-IPN cathode over the whole voltage range.

[Original manuscript]

Fig. 5 | f, R_{internal} and D_{Li^+} of the c-IPN (vs. control n-IPN) cathodes ($M/A = 65 \text{ mg cm}^{-2}$) as a function of the discharge voltage, which were estimated from the galvanostatic intermittent titration technique (GITT) results.

Supplementary Fig. 20 | a, Galvanostatic intermittent titration technique (GITT) profiles of the high- M/A ($= 65 \text{ mg cm}^{-2}$) cathodes (c-IPN vs. n-IPN) upon the repeated current stimuli at discharge current rate of 0.1 C ($= 1.35 \text{ mA cm}^{-2}$). **b**, GITT profiles showing the discharging step of the cells with the cathodes. The Li^+ diffusion coefficients (D_{Li^+}) were calculated using the following equation¹³:

$$(1) \quad D_{\text{Li}^+} = \frac{4}{\pi \Delta \tau} \left(\frac{m_{\text{B}} V_{\text{M}}}{M_{\text{B}} S} \right) \left(\frac{\Delta E_{\text{s}}}{\Delta E_{\text{t}}} \right)^2$$

where m_{B} is assigned to the mass of the electrode active material, S is the geometric area of the

electrode, M_B is the molar mass of the electrode material, V_M is the molar volume of the electrode material, and other parameters (Δt , ΔE_t and ΔE_s) in the equation are displayed in the GITT profiles shown above.

Supplementary Table 4 | Details of the calculation of the D_{Li^+} of the c-IPN (vs. n-IPN) cathodes.

	Δt (s)	M_B (g mol ⁻¹)	V_M (cm ³ mol ⁻¹)	$m_B S^{-1}$ (mg cm ⁻²)	ΔE_s (mV)	ΔE_t (mV)	D_{Li^+} (cm ² s ⁻¹)
n-IPN	3600	97.28	20.53	64.56	61.11	158.98	9.69 E-09
c-IPN	3600	97.28	20.53	64.48	51.54	86.52	2.32 E-08

The reviewer mentioned that the electrostatic repulsion of Li^+ by the c-IPN binder may hinder the charge transfer process. To address this issue, we monitored the change in the cell resistance by analyzing the EIS spectra during the charge/discharge cycling (**Supplementary Fig. 23** and **Supplementary Table 5**). At the 1st cycle, the c-IPN cathode showed a lower charge transfer resistance (R_{ct}) than the n-IPN cathode. This result demonstrates the advantageous effect of the c-IPN binder on the prevalence of freely mobile Li^+ and the uniform distribution of Li^+ concentration in the high C/A cathode (as shown in **Fig. 5**). The lower charge transfer resistance (R_{ct}) of the c-IPN cathode was still observed after 50th cycle, exhibiting the formation of stable CEI on NCM811 that could positively affect the cycling performance. This new result was added in the revised manuscript. The reviewer's valuable comment is highly appreciated, again.

[Revised manuscript]

Supplementary Fig. 23 | **a,b**, Nyquist plots of the Li-metal full cells (n-IPN cathode vs. c-IPN cathode at 100% SOC) at 1st (**a**), and 50th cycle (**b**). The symbols and solid lines represent experimental data and fitted curves, respectively. **c**, Equivalent circuits used to fit the curves¹⁴. The cathodes with an M/A of 65.0 mg cm⁻² (corresponding to C/A = 13.5 mAh cm⁻²) were assembled with Li-metal anodes (C/A = 20.0 mAh cm⁻², N/P ratio = 1.5).

Supplementary Table 5 | Analysis of the equivalent circuit elements (obtained from the Nyquist plots in **Supplementary Fig. 23**) as a function of cycle number.

Cathode	Impedance	1 st cycle	50 th cycle
	(ohm cm ²)		
n-IPN	R_{film}	9.3	79.8
	R_{ct}	17.0	47.0
c-IPN	R_{film}	9.2	23.6
	R_{ct}	4.0	33.6

[Revised manuscript]

“The effect of the c-IPN cathodes on the cycling performance of Li-metal full cells (cathodes ($C/A = 13.5 \text{ mAh cm}^{-2}$)||Li-metal anodes($C/A = 20.0 \text{ mAh cm}^{-2}$), N/P ratio = 1.5) was investigated. The cell with the c-IPN cathode exhibited a higher initial C_{sp} ($\sim 210 \text{ mAh g}_{\text{NCM811}}^{-1}$) than that with the control n-IPN cathode ($\sim 185 \text{ mAh g}_{\text{NCM811}}^{-1}$) (**Supplementary Fig. 22**). Moreover, under this constrained cell condition, the c-IPN cathode exhibited a higher cycling retention than the control cathode (capacity retention after 100 cycles = 82% vs. 16% for the control n-IPN cathode) (**Fig. 6a**). To further elucidate the effect of the c-IPN cathode on the cycling performance, we analyzed the EIS spectra of the cells. The c-IPN cathode showed a lower cell resistance than the n-IPN cathode at the 1st cycle (**Supplementary Fig. 23 and Supplementary Table 5**). This result verifies the advantageous effect of the c-IPN binder on the prevalence of freely mobile Li^+ and the uniform distribution of Li^+ concentration in the high C/A cathode (as shown in **Fig. 5**). The lower cell resistance of the c-IPN cathode was still observed after the 50th cycle, exhibiting the formation of stable CEI on NCM811 that could positively affect the cycling performance.”

To better comprehend the Li-ion transfer and improve Li transport, it is crucial to have a clear understanding of the microstructure of thick cathodes. For instance, the porosity of the electrodes plays a crucial role in determining the performance of the battery. Unfortunately, there is no mention of either the porosity or areal density in the current description, and it would be beneficial to include this information.

→ Thank you so much for the reviewer’s insightful comments. As the reviewer pointed out, the porosity of electrodes crucially affects the electrochemical performance of the battery. We already provided the electrode density in the original manuscript. Please see the paragraph below.

[Original manuscript]

“In addition to the aforementioned high-M/A, the c-IPN electrode exhibited a constant electrode density of 2.8 g cc^{-1} over the entire M/A range (**Fig. 3g**), which exceeded those of previously reported high-M/A electrodes (**Fig. 3h**).”

To ensure a fair comparison of the electrochemical performance between the n-IPN and the c-IPN electrodes, the electrode density of the electrodes was set to the same value. To clearly address this issue, an additional description was included in the revised manuscript.

[Revised manuscript]

Fig. 3 | **g**, Thickness and electrode density (ρ) of the c-IPN electrode as a function of M/A after the roll pressing. **h**, Electrode density of various electrodes as a function of C/A: c-IPN electrode vs. previously reported high-M/A electrodes.

“Fabrication of c-IPN electrodes

The electrode slurries were prepared with a composition ratio of $\text{LiNi}_{0.8}\text{Co}_{0.1}\text{Mn}_{0.1}\text{O}_2$ (NCM811),

LG energy solution)/carbon black additive (Super C65)/binder = 92/3/5 (w/w/w). For the control PVDF electrode, polyvinylidene fluoride (PVDF, Solvay) was solely used as binder and dissolved in N-methyl-2-pyrrolidone (NMP, Aldrich). For the control n-IPN and c-IPN electrodes, PVDF/trimethylolpropane triacrylate (TMPTA) (= 3/2 (w/w)) and PVDF/TMPTA/VAI-TFSI (= 3/0.7/1.3 (w/w/w)) were dissolved in NMP, respectively, in which benzoyl peroxide (BPO, 1 wt%) was used as a thermal initiator. The electrodes were fabricated by casting the electrode slurry on an Al current collector. The casted electrode slurry was dried at 100°C for 1h and followed by roll-pressing at 90°C. **The electrode density of the electrodes examined herein was set at 2.8 g cc⁻¹ for a fair comparison.** The electrodes were vacuum-dried at 120°C for 12 h before the cell assembly.”

Another improvement mechanism proposed was proposed as follows; "Consequently, PF6, which is adjacent to the c-IPN binder, was vulnerable to decomposition, thus promoting the formation of PF6-derived LiF-rich CEI layers." LiF enriched CEI layer may not positively impact the conductivity of CEI due to its low Li-ion conductivity. However, electrochemical data did not support this phenomenon and EIS fitting did not consider charge transfer impedance at all (SI Fig. 17).

→ Thank you so much for the reviewer’s insightful comments. As the reviewer mentioned, the inorganic (i.e., LiF)-rich CEI layer may not positively impact the conductivity of the CEI due to its low Li⁺ conductivity (σ_{Li^+}). However, when the inorganic-rich CEI layer has small thickness, the ionic conductance ($G \propto \frac{\sigma}{l}$, not ionic conductivity) through the thin inorganic-rich CEI layer is not significantly impaired. A similar discussion on the inorganic-rich CEI layer was reported in a previous paper (ref.: C. Wang et al., *Joule* **3**, 2550 (2019)).

the CEI contains a large amount of LiF from the fluorinated electrolyte. In addition, the presence of $\text{Li}_x\text{PO}_y\text{F}_z$ signal might be from the absorbed LiPF_6 salts and the decomposition of LiPF_6 in both electrolytes. Although the bulk LiF is highly resistive to Li-ion conduction, the *in-situ* formed LiF-rich CEI layer did not reduce the ion transport kinetics, due to (1) its small thickness (Figure 4D) and close contacting with LNO, also (2) much lower energy barrier for Li^+ surface diffusion (0.17 eV).^{39,44}

Fig. A captured image from a research article published in *Joule* **3**, 2550 (2019), “Designing In-Situ-Formed Interphases Enables Highly Reversible Cobalt-Free LiNiO_2 Cathode for Li-ion and Li-metal Batteries”.

In response to the reviewer’s comment, we monitored the change in the cell resistance by analyzing the EIS spectra during the charge/discharge cycling (**Supplementary Fig. 23** and **Supplementary Table 5**). The corresponding equivalent circuit model was taken from previous studies (ref.: J. B. Goodenough et al., *J. Electrochem. Soc.* **132**, 1521 (1985), D. Aurbach et al., *J. Power Sources* **89**, 206 (2020) and B. H. Farnum et al., *ACS Appl. Energy Mater.* **3**, 66 (2020)). At the 1st cycle, the c-IPN cathode showed a lower cell resistance (consisting of the CEI resistance (R_{film}) and charge transfer resistance (R_{ct})). This result verifies the advantageous effect of the c-IPN binder on the prevalence of freely mobile Li^+ and uniform distribution of Li^+ concentration in the high C/A cathode (as shown in **Fig. 5**). The lower cell resistance of the c-IPN cathode was still observed after the 50th cycle, exhibiting the formation of stable CEI on NCM811 that could beneficially affect the cycling performance. It is known that the inorganic-rich CEI layer tends to alleviate undesired interfacial side reactions with electrolytes, thus positively contributing to the cycle life (ref.: X. Qiu et al., *Nat. Commun.* **11**, 3629 (2020) and C. Wang et al., *Nat. Nanotech.* **13**, 715 (2018)). This new result was added in the revised manuscript. The reviewer’s valuable comment is highly appreciated, again.

[Original manuscript]

Fig. 7 | Structural characterization of the cycled NCM811.

a,b, Cross-sectional SEM images of the cycled NCM811 in the n-IPN (**a**) and c-IPN (**b**) cathodes. **c-f**, High-resolution transmission electron microscopy (HR-TEM) images of the cycled NCM811 in the n-IPN (**c,e**) and c-IPN (**d,f**) cathodes. **g,h**, Corresponding fast Fourier transform (FFT) patterns of the cycled NCM811 in the n-IPN (**g**) and c-IPN (**h**) cathodes.

[Revised manuscript]

Supplementary Fig. 23 | a,b, Nyquist plots of the Li-metal full cells (n-IPN cathode vs. c-IPN cathode at 100% SOC) at 1st (a), and 50th cycle (b). The symbols and solid lines represent experimental data and fitted curves, respectively. **c,** Equivalent circuits used to fit the curves¹⁴. The cathodes with an M/A of 65.0 mg cm⁻² (corresponding to C/A = 13.5 mAh cm⁻²) were assembled with Li-metal anodes (C/A = 20.0 mAh cm⁻², N/P ratio = 1.5).

Supplementary Table 5 | Analysis of the equivalent circuit elements (obtained from the Nyquist plots in **Supplementary Fig. 23**) as a function of cycle number.

Cathode	Impedance	1 st cycle	50 th cycle
	(ohm cm ²)		
n-IPN	R_{film}	9.3	79.8
	R_{ct}	17.0	47.0
c-IPN	R_{film}	9.2	23.6
	R_{ct}	4.0	33.6

[Revised manuscript]

“The effect of the c-IPN cathodes on the cycling performance of Li-metal full cells (cathodes ($C/A = 13.5 \text{ mAh cm}^{-2}$)||Li-metal anodes($C/A = 20.0 \text{ mAh cm}^{-2}$), N/P ratio = 1.5) was investigated. The cell with the c-IPN cathode exhibited a higher initial C_{sp} ($\sim 210 \text{ mAh g}_{\text{NCM811}}^{-1}$) than that with the control n-IPN cathode ($\sim 185 \text{ mAh g}_{\text{NCM811}}^{-1}$) (**Supplementary Fig. 22**). Moreover, under this constrained cell condition, the c-IPN cathode exhibited a higher cycling retention than the control cathode (capacity retention after 100 cycles = 82% vs. 16% for the control n-IPN cathode) (**Fig. 6a**). To further elucidate the effect of the c-IPN cathode on the cycling performance, we analyzed the EIS spectra of the cells. The c-IPN cathode showed a lower cell resistance than the n-IPN cathode at the 1st cycle (**Supplementary Fig. 23 and Supplementary Table 5**). This result verifies the advantageous effect of the c-IPN binder on the prevalence of freely mobile Li^+ and the uniform distribution of Li^+ concentration in the high C/A cathode (as shown in **Fig. 5**). The lower cell resistance of the c-IPN cathode was still observed after the 50th cycle, exhibiting the formation of stable CEI on NCM811 that could positively affect the cycling performance.

To investigate the CEI formed on NCM811, the NCM811 in the cycled cathodes were analyzed using X-ray photoelectron spectroscopy (XPS) depth profiles as a function of the etching time. The control n-IPN cathode exhibited a large content of organic elements (reflected by C and O) (**Fig. 6b**), which mostly originated from the decomposition of carbonate solvents in the electrolytes. In contrast, the c-IPN cathode exhibited a higher content of inorganic elements (such as Li and F) over a broad range of etching time, combined with the formation of LiF-rich components (**Supplementary Fig. 24**). It is known that the LiF-rich CEI layer tends to alleviate undesired interfacial side reactions with electrolytes, thus positively contributing to the cycle life^{32,33}. This unique CEI structure of the c-IPN cathode was further characterized using the time-of-flight secondary ion mass spectrometry (TOF-SIMS) depth profiling (**Supplementary Fig. 25**), which is consistent with the result of LiF-rich CEI layer formed in the LiPF_6 -based concentrated electrolytes³³.”

Overall, I observed that there are scattered assumptions regarding the improvement mechanism, which are not cohesively linked to each other. Additionally, the proposed mechanisms lack strong support from the electrochemical data.

→ Thank you so much for the reviewer's valuable comments. As described in the responses above, we've tried our best to address the reviewer's comments on the mechanism of improvement. Moreover, additional results have been included in the revised manuscript to further support the proposed mechanisms, including **Supplementary Figs. 1, 2, 4, 18, 23** and **Supplementary Table 5**. The reviewer's valuable comments are highly appreciated, again.

REVIEWERS' COMMENTS

Reviewer #2 (Remarks to the Author):

The authors responded diligently to the reviewers' questions and revised the manuscript accordingly, conducting additional characterization as requested. The concepts now sound well-explained and coherent. Therefore, I would like to recommend accepting the paper for publication.